# A Proximal Stochastic Gradient Method for Doubly-regularized Spectral Risk Minimization

## Abstract

Spectral risk minimization (SRM) is an important category of distributionally robust optimization. Recent works in this field elaborate on either distribution shift regularization (DSR) on the spectrum or non-differentiable regularization (NDR) on the parameters. However, few methods can simultaneously handle double regularization. The main difficulty lies in suppressing the bias and variance of the stochastic gradient when double regularization is present. To solve this problem, we develop a novel proximal stochastic gradient method (PSG-SRM) that simultaneously handles double regularization, reduces bias and variance along with iterations, and achieves linear convergence. It has lower computational complexity than two state-of-the-art methods that handle DSR or NDR separately. Experimental results indicate that it achieves competitive performance in both regression and classification tasks, and shows stable performance with respect to randomness.

## 1 Introduction

Distributionally robust optimization (DRO) is a general machine learning framework that can be formulated as follows without loss of generality:

$$\min_{w\in\mathbb{R}^d} \max_{\mathscr{Q}\in\mathcal{Q}}\{\mathbb{E}_{X\sim\mathscr{Q}}[\ell(w,X)] - \nu D(\mathscr{Q}\|P_n)\}, \tag{1}$$

where $w \in \mathbb{R}^d$ is a set of parameters with size $d$, $P_n$ is the empirical distribution of a training set $\{x_i\}_{i=1}^n$ with size $n$, and $\ell : \mathbb{R}^d \times \mathcal{X} \mapsto \mathbb{R}$ is a loss function. $\nu D(\mathscr{Q}\|P_n)$ is the **distribution shift regularization (DSR)** that measures the price for the test distribution $\mathscr{Q}$ to drift around an uncertainty set of probability measures $\mathcal{Q}$, $D(\mathscr{Q}\|P_n)$ is the divergence (e.g., the $\chi^2$ or K-L divergence) of $\mathscr{Q}$ from $P_n$ on the training set, and $\nu \geqslant 0$ is the shift cost hyperparameter. DRO tries to minimize the expected loss with respect to (w.r.t.) the most adversarial distribution shift in $\mathcal{Q}$ at a price of $\nu D(\mathscr{Q}\|P_n)$. This framework includes both supervised and unsupervised learning according to the data structure of $X$ and the specific form of $\ell$.

DRO has inspired a wide range of recent works in learning theory and applications, such as statistical learning (Duchi & Namkoong, 2021), reinforcement learning (Kumar et al., 2020), federated learning (Pillutla et al., 2024), language models (Liu et al., 2021), and robotics (Sharma et al., 2020). In practice, the probability measure $\mathscr{Q}$ can be parameterized by a set of weights $q \in \mathcal{P} := \{p \in \mathbb{R}^n : p \geqslant 0_n, p^\top 1_n = 1\}$, where $\mathcal{P}$ denotes the $n$-dimensional simplex (Lai et al., 2018; 2020; Lin et al., 2024a), $0_n$ and $1_n$ denote the $n$-dimensional zero and one vectors, respectively. Accordingly, the empirical distribution $P_n$ is represented by the uniform weights $1_n/n$. Then (1) can be reformulated as the following specific form (Mehta et al., 2024):

$$\min_{w\in\mathbb{R}^d} \mathcal{R}_\mathcal{P}(\ell(w)) \text{ for } \mathcal{R}_\mathcal{P}(l) := \max_{q\in\mathcal{P}}\left\{\sum_{i=1}^n q_i l_i - \nu D(q\|1_n/n)\right\}, \tag{2}$$

where $\ell(w) = (\ell_1(w), \cdots, \ell_n(w)) \in \mathbb{R}^n$ with $\ell_i(w) = \ell(w, x_i)$ being the loss for the $i$-th training sample. The different font $l$ indicates a table of losses that are maintained while computing $l \approx \ell(w)$ for the current iterate $w$, which will be explained in Appendix A.3.

Spectral risk minimization (SRM) is an important category of DRO that takes the superquantile or extremile (Daouia et al., 2019) as the objective function, such as the conditional value-at-risk (CVaR

Figure 1: Diagram of doubly-regularized SRM model. It takes advantage of both DSR and NDR that not only facilitates complicated regularizations but also achieves linear convergence.

(Rockafellar & Uryasev, 2000; Lin et al., 2024b), an important metric in risk measure), expected shortfall, average top-$k$ loss, and exponential spectral risk measure (ESRM). In general, SRM sorts $\ell_1(w), \cdots, \ell_n(w)$ in an ascending order $\ell_{(1)}(w) \leqslant \cdots \leqslant \ell_{(n)}(w)$, and constructs any vector $\sigma := (\sigma_1, \cdots, \sigma_n) \in \mathcal{P}$ such that $\sigma_1 \leqslant \cdots \leqslant \sigma_n$. Then $\sum_{i=1}^n \sigma_i \ell_{(i)}(w)$ is the maximum of $\sum_{i=1}^n q_i \ell_i(w)$ when $q$ is constrained in the uncertainty set $\mathcal{P}(\sigma) := \text{ConvexHull}\{\text{permutations of } \sigma\}$. Hence SRM can be specified by using $\mathcal{R}_{\mathcal{P}(\sigma)} =: \mathcal{R}_\sigma$ in (2).

Recently, a new bias and variance reduction approach *PROSPECT* (Mehta et al., 2024) has been proposed for the following strongly convex SRM model:

$$\min_{w \in \mathbb{R}^d} f(w) := \mathcal{R}_\sigma(\ell(w)), \quad \text{each } \ell_i(w) \text{ is } \mu\text{-strongly convex with } \mu > 0. \tag{3}$$

It exploits the $\mu$-strong convexity of $f$ to reduce the bias and variance of the gradient estimation and achieve linear convergence. Another new method *SOREL* (Ge & Jiang, 2025) replaces the DSR in (3) by a $\mu$-strongly convex **non-differentiable regularization (NDR)**:

$$\min_{w \in \mathbb{R}^d} \{\sum_{i=1}^n \sigma_i \ell_{(i)}(w) + g(w)\}, \quad g(w) \text{ is } \mu\text{-strongly convex with } \mu > 0. \tag{4}$$

This strategy enables SOREL to employ the elastic net regularization (Zou & Hastie, 2005) for sparse learning. However, since it drops the DSR, it loses some distributional robustness and only obtains sublinear convergence.

Since NDRs are widely-used to handle complicated tasks, such as out-of-distribution generalization (Lai & Wang, 2024; Wang et al., 2025), sparse learning (Lin et al., 2024a;b), and singularity problems (Lai et al., 2024; 2025), it motivates us to unify NDR and DSR into the following **doubly-regularized SRM model**:

$$\min_{w \in \mathbb{R}^d} \left\{ F(w) := f(w) + g(w) = \max_{q \in \mathcal{P}} \{\sum_{i=1}^n q_i \ell_i(w) - \underbrace{\nu D(q \| 1_n/n)}_{DSR}\} + \underbrace{g(w)}_{NDR} \right\}, \tag{5}$$

where each $\ell_i(w)$ is $\mu$-strongly convex, and $g : \mathbb{R}^d \mapsto \mathbb{R} \cup \{\pm\infty\}$ is a proper, lower semicontinuous, and (not necessarily strongly) convex NDR. The definitions of these terms are provided in Appendix A.1. This model not only enjoys the distributional robustness and linear convergence brought by the DSR, but also allows for more kinds of NDRs than (4). For example, the commonly-used $\ell_1$ regularization $g(w) = \lambda\|w\|_1$ with $\lambda \geqslant 0$ cannot be used in (4) due to the lack of strong convexity, but it can be used in (5). The diagram of the proposed doubly-regularized SRM model is shown in Figure 1. Equations (1) to (5) outline a specific development path in this field: (1) DSR $\rightarrow$ (2) SRM+DSR $\rightarrow$ (3) SRM+DSR with strongly convex loss $\rightarrow$ (4) SRM+NDR with strongly convex regularizer $\rightarrow$ (5) SRM+DSR+NDR (ours).

Neat as it is, (5) **is difficult to solve** for several reasons. **1.** Bias and variance reduction algorithms like PROSPECT (Mehta et al., 2024) can handle the DSR but cannot handle NDRs. **2.** Standard proximal (stochastic) gradient methods like SOREL can handle NDRs but cannot handle the DSR. **3.** The convergence results involving the NDR, DSR and bias & variance reduction are difficult to establish, because the bias and variance reduction mechanism only dominates $f$ but not $g$. To overcome these difficulties, we develop a novel proximal stochastic gradient method for (5), named *PSG-SRM*. **Our main contributions** can be summarized as follows:

**1.** We skillfully exploit the non-expansiveness of proximal mapping to preserve the bias and variance reduction mechanism for the double regularization in (5), yielding a descent property of PSG-SRM.

**2.** We develop a variance control scheme for the stochastic gradient, so that this variance can be reduced in the iterative algorithm. Moreover, this variance control scheme accelerates the proximal gradient descent step, which plays a key role in achieving linear convergence.

**3.** We develop an appropriate Lyapunov function for PSG-SRM to achieve linear convergence, which is better than sublinear convergence that a standard proximal (stochastic) gradient method can achieve (like SOREL). Moreover, the computational complexity of PSG-SRM is even lower than PROSPECT when the size of training set $n$ is also taken into consideration.

To better highlight our contributions, we summarize the requirements, functions, and computational complexities of the three algorithms for SRM in Table 1. PSG-SRM solves the most general problem with the lowest computational complexity among the three.

Table 1: Comparisons between different algorithms for spectral risk minimization. $n$ and $\varepsilon$ denote the number of training samples and the convergence tolerance, respectively. The computational complexity of SOREL is non-deterministic w.r.t. $n$.

| Method | Fidelity | DSR | NDR | Computational Complexity |
|---|---|---|---|---|
| PROSPECT | | Available | Unavailable | $\mathcal{O}\left(n \cdot \ln\left(\frac{(C_3' + C_4'n^2 + C_1'n)C_2'}{\varepsilon}\right)\right)$ |
| SOREL | $\mu$-strongly Convex, $G$-Lipschitz Continuous, $L_\ell$-smooth (All Three) | Unavailable | $\mu$-strongly Convex | $\tilde{\mathcal{O}}\left(\frac{1}{\sqrt{\varepsilon}}\right)$ |
| **PSG-SRM (ours)** | | Available | Convex | $\mathcal{O}\left(n \cdot \ln\left(\frac{(C_3 + C_1\mu n)C_2}{\varepsilon}\right)\right)$ |

## 2 PRELIMINARIES AND RELATED WORKS

Without loss of generality, we review some basic concepts in a way that best fits the context of this paper. A general stochastic gradient descent (SGD) approach aims to construct an estimate $v_i$ for the gradient $\nabla f(w)$ in (3) with only one sample $i$. This scheme does not need to wait and aggregate the full batch of gradients w.r.t. all the samples, which is suitable for general online learning or stream processing scenarios. Besides, it is empirically faster than the full-batch gradient descent approach at the early stage of the iteration in reducing the loss function. When $i$ is uniformly sampled from $[n] := \{1, 2, \cdots, n\}$ in each update iteration, $v_i \to \nabla f(w)$ holds ideally as the number of iterations increases. To achieve this, it is crucial to control the following expected squared error, which can be decomposed into the bias and variance terms:

$$\mathbb{E}[\|v_i - \nabla f(w)\|_2^2] = \underbrace{\|\mathbb{E}[v_i] - \nabla f(w)\|_2^2}_{\text{bias}} + \underbrace{\mathbb{E}[\|v_i - \mathbb{E}[v_i]\|_2^2]}_{\text{variance}}. \tag{6}$$

A suitable algorithm PROSPECT (Mehta et al., 2024) is developed to simultaneously reduce the above bias and variance along with the iterative process.

### 2.1 BIAS REDUCTION

The gradient $\nabla f(w) = \nabla_w \mathcal{R}_\sigma(\ell(w))$ is actually computed by the chain rule $(\nabla \ell(w))^\top \nabla_l \mathcal{R}_\sigma(l)$ provided $\ell(w)$ and $\mathcal{R}_\sigma(l)$ satisfy certain smoothness conditions, where $\nabla_l \mathcal{R}_\sigma(l)$ can be computed by Danskin's theorem (Bertsekas, 1999; Mehta et al., 2024):

$$\nabla_l \mathcal{R}_\sigma(l) = \underset{q \in \mathcal{P}(\sigma)}{\arg\max} \left\{ \sum_{i=1}^{n} q_i l_i - \nu D(q\|1_n/n) \right\}. \tag{7}$$

Details are provided in Appendix A.2. Denote $q^l := \nabla_l \mathcal{R}_\sigma(l)$, since it is actually a set of weights in $\mathcal{P}(\sigma)$. Then $\nabla_w \mathcal{R}_\sigma(\ell(w)) = \sum_{i=1}^{n} q_i^{\ell(w)} \nabla \ell_i(w)$. In practice, one has to fix $l \approx \ell(w)$ and then solve (7) for $q^l$ via its dual problem, which is illustrated in Appendix A.3. Lemma 6 indicates that the mapping $l \mapsto q^l$ is Lipschitz continuous and thus $\sum_{i=1}^{n} q_i^l \nabla \ell_i(w) \approx \sum_{i=1}^{n} q_i^{\ell(w)} \nabla \ell_i(w)$. Hence when $i$ is uniformly sampled from $[n]$, we have $\mathbb{E}[nq_i^l \nabla \ell_i(w)] \approx \nabla_w \mathcal{R}_\sigma(\ell(w))$. This forms an asymptotically unbiased gradient estimate.

## 2.2 VARIANCE REDUCTION

As for variance reduction, the control variate method is a widely-used approach in mathematical statistics. Suppose there is an estimator (not necessarily consistent) $\hat{a}$ of $a \in \mathbb{R}^d$, then we can exploit another estimator $\hat{b}$ in the same probability space with a known expectation $\mathbb{E}[\hat{b}] = b \in \mathbb{R}^d$ to construct a new estimator that reduces variance. Specifically, if $\mathbb{E}[(\hat{a} - a)^\top (\hat{b} - b)] > 0$, then the improved estimator can be $\tilde{a} := \hat{a} - \gamma(\hat{b} - b)$ with $\gamma > 0$ (or $\tilde{a} := \hat{a} + \gamma(\hat{b} - b)$ if $\mathbb{E}[(\hat{a} - a)^\top (\hat{b} - b)] < 0$). This can be seen by

$$\frac{\mathbb{E}[\|\hat{a} - \gamma(\hat{b} - b) - a\|_2^2]}{\mathbb{E}[\|\hat{a} - a\|_2^2]} = 1 - 2\gamma \frac{\mathbb{E}[(\hat{a} - a)^\top (\hat{b} - b)]}{\mathbb{E}[\|\hat{a} - a\|_2^2]} + o(\gamma) < 1. \tag{8}$$

In the literature, we let $\hat{a} = nq_i^l \nabla \ell_i(w)$ and $\hat{b} = n\rho_i \nabla \ell_i(z)$. With appropriate designs, $\rho - q^{\ell(w)} \to 0$ and $\nabla \ell(z) - \nabla \ell(w) \to 0$ as the number of iterations increases. Therefore, $\hat{b} - \hat{a} \to 0$. By setting $\gamma = 1$, the stochastic gradient can be set as

$$\hat{a} - \hat{b} + b \approx nq_i^l \nabla \ell_i(w) - n\rho_i \nabla \ell_i(z) + \sum_{j=1}^{n} \rho_j \nabla \ell_j(z) =: v_i. \tag{9}$$

Suppose $v^{(t)}$ is the sampled stochastic gradient at the $t$-th iteration, then the variance reduction step aims to asymptotically suppress its variance $\mathrm{Var}(v^{(t)}) \to 0$ as $t \to \infty$. Such a variance reduction scheme accelerates the gradient descent step and leads to linear convergence. This scheme even allows for a differentiable convex regularizer $g(w)$ because such a regularizer can be absorbed in the loss function $\ell(w)$. However, when $g(w)$ is an NDR, this variance reduction scheme fails because the gradient $\nabla g(w)$ does not exist and cannot be absorbed in $\nabla \ell(w)$. Hence the PROSPECT algorithm cannot handle an NDR.

## 2.3 PRIMAL-DUAL APPROACH

As for SRM with only NDR, an intuitive approach is to consider the weights $q$ as dual variables and reformulate (4) as a primal-dual optimization problem (Ge & Jiang, 2025):

$$\min_{w \in \mathbb{R}^d} \max_{q \in \mathcal{P}(\sigma)} \{\mathcal{L}(w, q) := \sum_{i=1}^{n} q_i \ell_i(w) + g(w)\},$$

$$\begin{cases} q^{(t+1)} = \mathrm{argmax}_{q \in \mathcal{P}(\sigma)} \{\sum_{i=1}^{n} q_i \ell_i(w^{(t)}) - \frac{1}{\eta^{(t)}} \|q - q^{(t)}\|_2^2\}, \\ w^{(t+1)} = \mathrm{argmin}_{w \in \mathbb{R}^d} \{\sum_{i=1}^{n} q_i^{(t+1)} \ell_i(w) + g(w) + \frac{1}{\vartheta^{(t)}} \|w - w^{(t)}\|_2^2\}, \end{cases} \tag{10}$$

where $\eta^{(t)}$ and $\vartheta^{(t)}$ are the dual and primal learning rates, respectively. In both primal and dual updates, proximal terms ($-\frac{1}{\eta^{(t)}} \|q - q^{(t)}\|_2^2$ and $\frac{1}{\vartheta^{(t)}} \|w - w^{(t)}\|_2^2$) are added to stabilize the trajectory of $q$ and facilitate strong convexity on the function $g + \frac{1}{\vartheta^{(t)}} \| \cdot - w^{(t)}\|_2^2$. By carefully setting $\vartheta^{(t)} = \mathcal{O}\left(\frac{1}{t}\right)$, the SOREL algorithm (Ge & Jiang, 2025) can achieve a near optimal sublinear rate of $\tilde{\mathcal{O}}\left(\frac{1}{\sqrt{\varepsilon}}\right)$ for a convergence tolerance of $\varepsilon > 0$. However, since it drops the DSR, it loses some distributional robustness and does not smooth the spectral risk, thus its convergence rate is lower than that of PROSPECT.

## 3 METHODOLOGY

To simultaneously improve distributional robustness, achieve high convergence rate, and allow for complicated regularizations, we propose to solve the doubly-regularized SRM model (5) with both DSR and NDR. In (5), each $\ell_i(w)$ is $\mu$-strongly convex, $G$-Lipschitz continuous, and $L_\ell$-smooth:

$$\mu\text{-strongly Convex:} \quad \ell_i(w_2) \geqslant \ell_i(w_1) + (\nabla \ell_i(w_1))^\top (w_2 - w_1) + \frac{\mu}{2} \|w_2 - w_1\|_2^2, \tag{11}$$

$$G\text{-Lipschitz Continuous:} \quad |\ell_i(w_2) - \ell_i(w_1)| \leqslant G\|w_2 - w_1\|_2, \tag{12}$$

$$L_\ell\text{-smooth:} \quad \|\nabla \ell_i(w_2) - \nabla \ell_i(w_1)\|_2 \leqslant L_\ell \|w_2 - w_1\|_2, \quad \forall w_1, w_2 \in \mathbb{R}^d. \tag{13}$$

$g : \mathbb{R}^d \mapsto \mathbb{R} \cup \{\pm\infty\}$ is a proper, lower semicontinuous, and (not necessarily strongly) convex NDR. The detailed explanations of these terms are provided in Appendix A.1.

### 3.1 LIPSCHITZ SMOOTHNESS OF $f$

It directly follows from the $G$-Lipschitz continuity and smoothness of $\ell$ that for any $w \in \mathbb{R}^d$,

$$\|\nabla \ell(w)\|_2 \leqslant \|\nabla \ell(w)\|_F = \sqrt{\mathrm{tr}((\nabla \ell(w))^\top \nabla \ell(w))} = \sqrt{\sum_{i=1}^n \|\nabla \ell_i(w)\|_2^2} \leqslant \sqrt{n} G. \qquad (14)$$

$f$ is also $\mu$-strongly convex since it is a convex combination of $\ell_i$. According to (28) and Lemma 6, $w \mapsto q^{opt}(\ell(w))$ is $L_{q \circ \ell} := \frac{G}{\sqrt{n} \alpha_n \nu}$-Lipschitz continuous (Lemma 7), where $\alpha_n$ denotes the strong convexity parameter associated with the generating function of the divergence $D(q \| 1_n / n)$ (see Appendix A.2 for details). Then we can obtain the Lipschitz smoothness of $f$.

**Lemma 1.** *For any* $w, w' \in \mathbb{R}^d$, $\|\nabla f(w) - \nabla f(w')\|_2 \leqslant L_f \|w - w'\|_2$ *for the constant* $L_f := M_q L_\ell + \sqrt{n} G L_{q \circ \ell}$, *where* $M_q := \sup_{w \in \mathbb{R}^d} \|q^{opt}(\ell(w))\|_2$. $M_q < \infty$ *because* $q$ *lies in a compact set* $\mathcal{P}(\sigma)$.

The proof is provided in Appendix A.5.

### 3.2 PROXIMAL STOCHASTIC GRADIENT FOR SPECTRAL RISK MINIMIZATION

To better understand the key mechanisms of the proposed PSG-SRM method, we present one iteration of the algorithm along with corresponding explanations in Algorithm 1.

---

**Algorithm 1:** PSG-SRM

---

**Require:** Initialize: $t = 0$, $z^{(t)} = w^{(t)} = 0$. Set learning rate $\eta = \frac{1}{(1+\delta)L_f}, \delta \in (0, 1)$,
    convergence tolerance $\varepsilon > 0$.
1: **for** $t = 0, 1, 2, \cdots, T - 1$ **do**
2:     Sample two independent indices: $i_t \sim \mathrm{Unif}([n]), j_t \sim \mathrm{Unif}([n])$.
3:     Compute stochastic gradient: $v^{(t)} = n q_{i_t}^{(t)} \nabla \ell_{i_t}(w^{(t)}) - (n \rho_{i_t}^{(t)} \nabla \ell_{i_t}(z_{i_t}^{(t)}) - \nabla \bar{\ell}^{(t)})$.
4:     Reduce bias: $\zeta_{j_t}^{(t+1)} = w^{(t)}$ and $\zeta_j^{(t+1)} = \zeta_j^{(t)}, \forall j \neq j_t$.
5:     Update $q^{(t+1)}$ by (7) with $l^{(t+1)} := \ell(\zeta^{(t+1)})$.
6:     Reduce variance: $z_{i_t}^{(t+1)} = w^{(t)}$ and $z_i^{(t+1)} = z_i^{(t)}, \forall i \neq i_t$.
7:     $\rho_{i_t}^{(t+1)} = q_{i_t}^{(t)}$ and $\rho_i^{(t+1)} = \rho_i^{(t)}, \forall i \neq i_t$.
8:     $\nabla \bar{\ell}^{(t+1)} = \sum_{i=1}^n \rho_i^{(t+1)} \nabla \ell_i(z_i^{(t+1)})$.
9:     Proximal stochastic gradient step:
       $w^{(t+1)} = \mathrm{prox}_{\eta g}(w^{(t)} - \eta v^{(t)}) := \mathrm{argmin}_{w \in \mathbb{R}^d} \{ \frac{1}{2} \|w - (w^{(t)} - \eta v^{(t)})\|_2^2 + \eta g(w) \}$.
10: **end for**
**Ensure:** $w^{(T)}$.

---

Besides, we also present the following three important intermediate variables, although they need not be computed in the iteration.

$$\begin{cases} h^{(t)} = \frac{1}{\eta}(w^{(t)} - w^{(t+1)}). & \text{Update Gap} \\ \Delta^{(t)} = v^{(t)} - \nabla f(w^{(t)}). & \text{Stochastic Gradient Gap} \\ \bar{w}^{(t+1)} = \mathrm{prox}_{\eta g}(w^{(t)} - \eta \sum_{i=1}^n \rho_i^{(t)} \nabla \ell_i(z_i^{(t)})). & \text{Full Proximal Gradient Step} \end{cases} \qquad (15)$$

Compared with the PROSPECT algorithm (Mehta et al., 2024), PSG-SRM needs to find a suitable learning rate $\eta$ according to the Lipschitz constant $L_f$ and implement the proximal stochastic gradient step. Due to this newly-added step, it is nontrivial to establish theoretical results for PSG-SRM, such as convergence properties. **The main difficulty lies in how to suppress the variance of the stochastic gradient gap $\Delta^{(t)}$ in the new scheme.**

To address the above difficulty, we first examine the mean of $\Delta^{(t)}$, and then bound the variance of $\Delta^{(t)}$, by the following lemma.

**Lemma 2.** *Let $w^*$ be an optimal point and $q^*$ be the corresponding optimal weights. Then*

$$\mathbb{E}_t[\Delta^{(t)}] = 0, \tag{16}$$

$$\mathbb{E}_t[\|\Delta^{(t)}\|_2^2] \leqslant 2Q^{(t)} + 2S^{(t)}, \tag{17}$$

*where $Q^{(t)}$ and $S^{(t)}$ are two types of errors:*

$$Q^{(t)} := \frac{1}{n}\sum_{i=1}^n \|nq_i^{(t)}\nabla\ell_i(w^{(t)}) - nq_i^*\nabla\ell_i(w^*)\|_2^2,$$

$$S^{(t)} := \frac{1}{n}\sum_{i=1}^n \|n\rho_i^{(t)}\nabla\ell_i(z_i^{(t)}) - nq_i^*\nabla\ell_i(w^*)\|_2^2. \tag{18}$$

*Besides, they have the following relationship:*

$$\mathbb{E}_t[S^{(t+1)}] := \frac{1}{n}Q^{(t)} + (1 - \frac{1}{n})S^{(t)}. \tag{19}$$

The proof is provided in Appendix A.6. (19) is a crucial equation that the conditional expectation of $S^{(t+1)}$ at iteration $t$ can be split into two terms with weights $\frac{1}{n}$ and $(1 - \frac{1}{n})$. By this means, part of the error in $S^{(t+1)}$ is transferred to $\frac{1}{n}Q^{(t)}$, which can be reduced by the descent property of PSG-SRM (Lemma 4). As for $(1 - \frac{1}{n})S^{(t)}$, it can be reduced with a factor $(1 - \frac{1}{n}) < 1$ at each iteration, which tends to zero in the end. This mechanism finally suppresses the variance of the stochastic gradient gap $\Delta^{(t)}$.

Next, we recall the following lemma for the descent property of a general proximal stochastic gradient step.

**Lemma 3** (Descent Property of General Proximal Stochastic Gradient (Xiao & Zhang, 2014))**.** *Let $\eta \in (0, \frac{1}{L_f}]$, then for any $w$,*

$$F(w) \geqslant F(w^{(t+1)}) + \langle h^{(t)}, w - w^{(t)}\rangle + \frac{\eta}{2}\|h^{(t)}\|_2^2 + \frac{\mu}{2}\|w - w^{(t)}\|_2^2 + \langle\Delta^{(t)}, w^{(t+1)} - w\rangle. \tag{20}$$

The proof is provided in Appendix A.7. Then by combining Lemma 2 and Lemma 3, we can obtain the descent property of PSG-SRM. **The key technique is to skillfully exploit the full proximal gradient step $\bar{w}^{(t+1)}$ in (15), which is independent of $i_t$, to separate the stochastic terms from the deterministic terms in (20).**

**Lemma 4** (Descent Property of PSG-SRM)**.**

$$\mathbb{E}_t[\|w^{(t+1)} - w^*\|_2^2] \leqslant (1 - \frac{\mu}{2})\|w^{(t)} - w^*\|_2^2 - 2\eta[\mathbb{E}_t[F(w^{(t+1)})] - F(w^*)] + 4\eta^2(Q^{(t)} + S^{(t)}), \tag{21}$$

*where $w^*$ is a solution to (5).*

The proof is provided in Appendix A.8. This lemma indicates that the descent property of PSG-SRM involves both of the argument gap $\|w^{(t+1)} - w^*\|_2$ and the function gap $[\mathbb{E}_t[F(w^{(t+1)})] - F(w^*)]$, which is substantially different from the descent property of PROSPECT (Lemma 11 in Mehta et al. 2024) that only involves the argument gap. The reason is that the strong convexity of $f$ yields the descent of argument gap, while the proximal gradient step yields the descent of the function gap. Besides, we skillfully exploit the non-expansiveness of proximal mapping $\text{prox}_{\eta g}$ when $\eta g$ is convex, in order to avoid expansion of the bias and variance.

### 3.3 CONVERGENCE ANALYSIS AND COMPUTATIONAL COST

To establish convergence results, it is crucial to develop an appropriate Lyapunov function, which exploits the descent property of both argument gap and function gap, as well as the error splitting in (19). We successfully develop such a Lyapunov function and establish a linear convergence result of PSG-SRM w.r.t. this Lyapunov function.

**Theorem 1** (Convergence Theorem). *Consider the following Lyapunov function*

$$V^{(t)} = \|w^{(t)} - w^*\|_2^2 + c_1 \cdot 4\eta^2 S^{(t)}. \tag{22}$$

*If the coefficient $c_1 > n$ and the learning rate $\eta \in (0, \sqrt{\frac{\mu}{8L_f(n+c_1)}})$, then*

$$\begin{cases} \mathbb{E}_t[V^{(t+1)}] \leqslant (1-\tau)V^{(t)} \\ \mathbb{E}[\|w^{(t)} - w^*\|_2^2] \leqslant (1-\tau)^t(1 + 8c_1n\eta^2(\|q^{(0)}\|_2^2 L_\ell^2 + G^2 L_{q\circ\ell}^2))\|w^{(0)} - w^*\|_2^2 \end{cases} \tag{23}$$

*holds for any*

$$\tau \in (0, \min\{\frac{\mu}{2} - 4\eta^2 L_f(n+c_1), \frac{1}{n} - \frac{1}{c_1}\}]. \tag{24}$$

*It indicates that both the Lyapunov function sequence $\{V^{(t)}\}_{t \geqslant 1}$ and the iterate sequence $\{w^{(t)}\}_{t \geqslant 1}$ converge linearly. By setting the coefficients and hyperparameters as follows:*

$$c_1 = 2n, \quad \eta = \frac{1}{2}\sqrt{\frac{\mu}{8L_f(n+c_1)}}, \quad \tau = \frac{1}{3n}, \tag{25}$$

*the number of iterations of PSG-SRM is $\mathcal{O}\left(n \cdot \ln\left(\frac{(C_3 + C_1\mu n)C_2}{\varepsilon}\right)\right)$ to achieve $\mathbb{E}[\|w^{(t)} - w^*\|_2^2] \leqslant \varepsilon$ for a convergence gap $\varepsilon > 0$, where $C_1, C_2, C_3 \geqslant 0$ are three constants independent of $n$ and $\varepsilon$.*

The proof is provided in Appendix A.9. The computational complexity of PSG-SRM is lower than that of PROSPECT $\mathcal{O}\left(n \cdot \ln\left(\frac{(C_3' + C_4'n^2 + C_1'n)C_2'}{\varepsilon}\right)\right)$, considering both $n$ and $\varepsilon$. Note that the standard proximal (stochastic) gradient method such as SOREL (Ge & Jiang, 2025) or (Xiao & Zhang, 2014) can only achieve a sublinear convergence result $\tilde{\mathcal{O}}\left(\frac{1}{\sqrt{\varepsilon}}\right)$ or $\mathcal{O}\left(\frac{1}{\varepsilon}\right)$ due to the non-differentiability of $g(w)$ and the non-contraction of $\text{prox}_{\eta g}$. But in PSG-SRM, the bias and variance reduction scheme is exploited to suppress $V^{(t)}$ in a faster way. Hence PSG-SRM not only reduces the bias and variance of gradient estimation, but also saves computational cost compared with the standard proximal (stochastic) gradient method.

As for the computational complexity of one iteration, the update of $q^{(t+1)}$ by $\zeta^{(t+1)}$ in Algorithm 1 involves a bubble sorting of the losses $l$ and a Pool Adjacent Violators (PAV) algorithm that solves (7). Since there is only one element of $l$ changes in one iteration, the bubble sorting of $l$ requires only $\mathcal{O}(n)$ operations. When $g(w) = \lambda\|w\|_1$ or other similar NDRs (e.g., the elastic net Zou & Hastie 2005), the proximal stochastic gradient step in Algorithm 1 has a closed-form solution with $\mathcal{O}(d)$ operations. Other steps in Algorithm 1 require either $\mathcal{O}(n)$ or $\mathcal{O}(d)$ operations. Hence the total computational complexity of one iteration is $\mathcal{O}(n + d)$, which is the same as that of PROSPECT, while PROSPECT cannot solve (5). The actual runtime results provided in Appendix B.6 also show that PSG-SRM achieves faster per-epoch computational speed than PROSPECT in most cases.

## 4 EXPERIMENTS

We perform regression and classification tasks to evaluate the effectiveness and robustness of PSG-SRM, compared with PROSPECT, LSVRG (Mehta et al., 2023), and SOREL. More implementation details and experimental results are provided in Appendix B.

### 4.1 EXPERIMENTAL SETTINGS

Optimization models under spectral risks with different types of regularizations where $g(w)$ is set to be $\lambda\|w\|_1$ (PSG-SRM), $\lambda\|w\|_1 + \frac{1}{2\tau}\|w - w^{(t)}\|_2^2$ (SOREL), $\frac{\lambda}{2}\|w\|_2^2$ (PROSPECT), or 0 (LSVRG) are trained. Note that only PSG-SRM can be used as the optimizer for $\ell_1$ regularization, while SOREL uses the elastic net regularization with an accelerated first-order momentum mechanism, which is a strongly-convex regularizer. PROSPECT can be used for the differentiable $\ell_2$ regularization, while LSVRG is used for non-regularization (i.e., $g(w) = 0$). Although each competitor can use its own optimization model to perform the tasks in the experiments, only PSG-SRM can solve the doubly-regularized SRM model (5) as a theoretically-sound approach.

Uncertainty sets generated by CVaR, extremile, and ESRM are considered in the experiments (see Appendix B.1). For regression tasks or multi-class classification tasks, the squared loss or the multi-nomial logistic loss is used for each sample, respectively (see Appendix B.2). The final objective function takes the following form

$$F(w) = \left\{ \max_{q \in \mathcal{P}(\sigma)} \sum_{i=1}^{n} q_i \ell_i(w) - \nu n \|q - 1_n/n\|_2^2 \right\} + g(w),$$

where the $\chi^2$-divergence is instantiated for $D(q\|1_n/n)$.

For LSVRG and SOREL, the epoch length is set to the training sample size. The number of training epochs for each data set follows the settings in (Mehta et al., 2024). All the hyperparameters including the shift cost $\nu$, the regularization strength $\lambda$, and the learning rate $\eta$ for LSVRG, PROSPECT, and SOREL are set according to their default criteria, while the hyperparameters for PSG-SRM are set according to Appendix B.4. All the experiments are repeated for 10 times with different random seeds drawn from $\{1, 2, \cdots, 10\}$, then the mean and standard deviation (STD) results are reported.

## 4.2 LEAST-SQUARES REGRESSION

Five benchmarks are used for the regression task under the squared loss: yacht ($n = 244$) (Tsanas & Xifara, 2012), energy ($n = 614$) (Baressi Šegota et al., 2019), concrete ($n = 824$) (Yeh, 2006), kin8nm ($n = 6553$) (Akujuobi & Zhang, 2017), and power ($n = 7654$) (Tüfekci, 2014). Table 2 presents the average squared losses on the test sets. PSG-SRM achieves the best performance in most cases, especially when the sample sizes are large, like energy, concrete, kin8nm, and power. These results indicate that the proposed doubly-regularized SRM model and the PSG-SRM algorithm are effective in these regression tasks. Moreover, PSG-SRM also achieves the most robust performance in nearly all the cases with the lowest STDs. In the situations where different methods achieve close means (like power), PSG-SRM further achieves the lowest STDs. Hence PSG-SRM is a stable algorithm w.r.t. randomness, which is an advantage as a stochastic method.

Table 2: Performance of different optimizers on regression benchmarks (mean± STD).

| Data Set | | SRM | PROSPECT | LSVRG | SOREL | PSG-SRM |
|---|---|---|---|---|---|---|
| yacht | CVaR | **0.1989**±1.2e-6 | 0.2009±1.4e-5 | 0.1993±1.4e-6 | 0.2021±**3.7e-16** |
| | Extremile | **0.1989**±1.0e-6 | 0.2010±1.2e-5 | 0.2001±1.1e-7 | 0.2023±**3.4e-16** |
| | ESRM | **0.2009**±8.0e-7 | 0.2056±1.6e-5 | 0.2012±1.6e-7 | 0.2039±**4.8e-14** |
| energy | CVaR | 0.0628±9.5e-7 | 0.0631±2.0e-4 | 0.0627±1.7e-6 | **0.0626**±2.2e-7 |
| | Extremile | 0.0628±9.4e-7 | 0.0627±3.1e-6 | 0.0628±2.4e-7 | **0.0626**±2.2e-7 |
| | ESRM | 0.0628±8.1e-7 | 0.0627±4.9e-6 | 0.0627±2.7e-7 | **0.0626**±2.2e-7 |
| concrete | CVaR | 0.2276±7.7e-6 | 0.2280±1.6e-4 | 0.2272 ±3.5e-6 | **0.2260**±2.5e-9 |
| | Extremile | 0.2276±7.7e-6 | 0.2279±5.6e-6 | 0.2272±2.0e-6 | **0.2260**±2.5e-9 |
| | ESRM | 0.2270±5.2e-6 | 0.2277±3.4e-6 | 0.2267 ±2.1e-6 | **0.2260**±2.5e-9 |
| kin8nm | CVaR | 0.2891±8.0e-9 | **0.2876**±1.2e-5 | 0.2877±3.1e-7 | 0.2877±**1.1e-5** |
| | Extremile | 0.2890±1.2e-7 | 0.2883 ±1.3e-4 | 0.2883±**7.3e-9** | **0.2876**±7.8e-9 |
| | ESRM | 0.2881±2.8e-9 | 0.2880 ±4.4e-6 | 0.2880 ±**7.2e-10** | **0.2876**±7.8e-9 |
| power | CVaR | 0.0369±2.3e-10 | 0.0378±1.3e-4 | 0.0370±7.5e-7 | **0.0369**±5.6e-11 |
| | Extremile | 0.0369±2.4e-10 | 0.0369±6.2e-5 | 0.0369±2.9e-7 | **0.0369**±5.6e-11 |
| | ESRM | 0.0369±2.2e-10 | 0.0369±7.9e-6 | 0.0369±7.0e-8 | **0.0369**±5.6e-11 |

Following the study of (Williamson & Menon, 2019), we examine how the compared optimizers improve fairness in the regression task. Adult Census (acsincome) is a common benchmark of regression tasks on fairness. It predicts the income of U.S. adults with data collected by the American Community Survey (Ding et al., 2021). The fairness is evaluated by the statistical parity score (SPC, Agarwal et al. 2019), which quantifies discrepancies in outcomes across different values of a protected attribute that is excluded from training. Specifically, denote $(X, Y, A)$ as a random (input, output, attribute) triplet, then a model $\mathcal{G}$ satisfies statistical parity if the conditional distribution of

$\mathcal{G}(X)$ over the predicted output under $A = a$ is equal for every value $a$. In other words, statistical parity measures the maximum gap between these distributions for different values of $a$, thus a statistical parity score closer to zero indicates higher fairness. In our task, the race and the gender attributes serve as the protected attributes (PA). Results are provided in Table 3. PSG-SRM achieves the lowest statistical parity scores in both spectral risks and both PAs. Specifically, PSG-SRM achieves 70.30% in CVaR and 70.14% in ESRM for the race PA, which are nearly 2% lower than the second best method SOREL. It also achieves 4.52% in CVaR and 4.76% in ESRM for the gender PA, which are nearly 0.2% lower than the second best method PROSPECT. Since SOREL and PROSPECT are two state-of-the-art methods corresponding to the NDR or DSR, respectively, the above results indicate that PSG-SRM achieves good fairness besides competitive regression performance as a method of double regularization.

Table 3: Fairness evaluation of different optimizers on `acsincome` (mean± STD). A lower value indicates higher fairness. Two protected attributes (PA) are considered: race and gender.

| PA | SRM | PROSPECT | LSVRG | SOREL | PSG-SRM |
|---|---|---|---|---|---|
| Race | CVaR | 0.7540±1.1e-16 | 0.7340±0.0000 | 0.7240±0.0000 | **0.7030**±1.8e-3 |
| | ESRM | 0.7340±0.0000 | 0.7300±1.1e-16 | 0.7280±0.0000 | **0.7014**±4.4e-3 |
| Gender | CVaR | 0.0495±0.0000 | 0.0510±8.0e-4 | 0.0514±0.0000 | **0.0452**±8.7e-4 |
| | ESRM | 0.0494±0.0000 | 0.0523±0.0000 | 0.0527±0.0000 | **0.0476**±1.9e-4 |

### 4.3 OUT-OF-DISTRIBUTION CLASSIFICATION

We also consider two classification benchmarks under distribution shifts: `amazon` and `iwildcam`. The `amazon` data set contains product reviews used to predict ratings from 1 to 5, with disjoint sets of reviewers in training and test (Ni et al., 2019). Following the procedure described in (Mehta et al., 2024), we extract review representations by a pretrained BERT model (Devlin et al., 2019) fine-tuned on a held-out subset of the training set for 2 epochs, then adopt the principal component analysis to subsequently reduce dimensionality. The `iwildcam` data set originates from the iWildCam image classification challenge (Beery et al., 2021) and consists of labeled wildlife images captured at various field sites. Distribution shifts arise from changes in camera angles, lighting conditions, locations, and other environmental factors. Deep features for `iwildcam` are extracted by a ResNet-50 model (He et al., 2016) pretrained on ImageNet (Russakovsky et al., 2015) without fine-tuning.

For both `amazon` and `iwildcam`, linear probe classifiers over frozen deep representations are trained with $n = 10000$ and $n = 20000$ examples, respectively. The extremile is used as the spectral risk type because it represents the most adverse situation in the out-of-distribution evaluation. The worst group test errors on `amazon` and the median group test errors on `iwildcam` for the compared optimizers are reported in Table 4. On `amazon`, PSG-SRM achieves the best performance among all the compared optimizers with the lowest worst group test error of 76.74%±1.1e-16, which shows the effectiveness of double regularization with both DSR and NDR. The second best method is PROSPECT, which also has the DSR. However, LSVRG and SOREL deteriorate substantially in performance, reaching test errors higher than 97%. The reason is that LSVRG may fail to converge in some cases, especially for small shift costs. SOREL does not have the DSR, hence it lacks robustness to such an extreme out-of-distribution test scenario. On `iwildcam`, PSG-SRM also shows the best out-of-distribution generalization performance with the lowest median group test error of 70.85%±0.0000, which is nearly 2.5% lower than SOREL and substantially lower than both PROSPECT and LSVRG. The results also indicate that LSVRG and SOREL have better performance in the median group test error than in the worst group test error.

Table 4: Out-of-distribution classification benchmarks on extremile (mean± STD). Performance is evaluated using the worst group test error on `amazon` and the median group test error on `iwildcam`. A lower value indicates better performance.

| Data Set | PROSPECT | LSVRG | SOREL | PSG-SRM |
|---|---|---|---|---|
| amazon | 0.7791 ±1.1e-16 | 0.9963±7.3e-3 | 0.9714±5.2e-2 | **0.7674±1.1e-16** |
| iwildcam | 0.7532±2.4e-3 | 0.7483±0.0000 | 0.7322±0.0000 | **0.7085±0.0000** |

## 5 CONCLUSION AND DISCUSSION

Spectral risk minimization (SRM) is an important methodology that has been widely applied in various fields of machine learning, such as the conditional value-at-risk, expected shortfall, and average top-$k$ loss. There are mainly two kinds of regularizations that can be imposed on SRM to achieve specific purposes. The distribution shift regularization (DSR) on the spectrum improves distributional robustness and the convergence rate, while the non-differentiable regularization (NDR) on the parameters facilitates complicated tasks such as out-of-distribution generalization and sparse learning.

In this study, we propose a doubly-regularized SRM with both DSR and NDR to exploit their advantages. To overcome the difficulties of handling this double regularization, we exploit the non-expansiveness of proximal mapping to develop a variance control scheme for the stochastic gradient, forming a novel proximal stochastic gradient method named PSG-SRM. Moreover, we develop an appropriate Lyapunov function for PSG-SRM to achieve linear convergence, which is superior than the sublinear convergence of a standard proximal (stochastic) gradient method. When taking the sample size into consideration, PSG-SRM also achieves lower computational complexity than two state-of-the-art solvers that handle DSR or NDR separately.

Experimental results show that PSG-SRM achieves competitive performance in regressions, fairness, and out-of-distribution classifications. It also shows stable performance with respect to different random seeds, which is an advantage as a stochastic method. Future works may lie in developing new PSG algorithms for the nonconvex setting of the loss function or the regularization term, or both. Other possible approaches are finding new effective double regularization schemes or developing new distributionally robust optimization models.

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

# A  THEORETICAL DETAILS

## A.1  DEFINITIONS OF BASIC TERMS

First, we provide some definitions of basic terms used in this paper. Note that we consider the extended real space $\bar{\mathbb{R}} := \mathbb{R} \cup \{\pm\infty\}$ as a general range of function values.

**Definition A1** (Convex, Strictly Convex, and Strongly Convex Functions)**.** A function $h : \mathbb{R}^n \mapsto \bar{\mathbb{R}}$ is called convex if for any $x, y \in \mathbb{R}^n$ and any $\alpha \in (0, 1)$,

$$h(\alpha x + (1 - \alpha)y) \leqslant \alpha h(x) + (1 - \alpha)h(y).$$

Furthermore, $h$ is strictly convex if for any $x \neq y$,

$$h(\alpha x + (1 - \alpha)y) < \alpha h(x) + (1 - \alpha)h(y).$$

In addition, $h$ is $\mu$-strongly convex ($\mu > 0$) if and only if $h - \frac{\mu}{2}\| \cdot \|_2^2$ is convex.

It can be seen that "$\mu$-strong convexity $\Rightarrow$ strict convexity $\Rightarrow$ ordinary convexity" but the converse cannot hold.

**Definition A2** (Proper and Lower Semicontinuous Functions)**.** $h : \mathbb{R}^n \mapsto \bar{\mathbb{R}}$ is a proper function if $\{x \in \mathbb{R}^n : h(x) < +\infty\} \neq \varnothing$ and $-\infty \notin h(\mathbb{R}^n)$. In brief, $h$ has a nonempty domain and never takes $-\infty$ as a function value.

$h : \mathbb{R}^n \mapsto \bar{\mathbb{R}}$ is lower semicontinuous at a point $x \in \mathbb{R}^n$ if

$$\liminf_{y \neq x, \, y \to x} h(y) := \lim_{\delta \to 0^+} \left( \inf_{y \in B(x;\delta)} h(y) \right) \geqslant h(x),$$

where $B(x; \delta)$ is a neighborhood of $x$ with radius $\delta$. Moreover, if $h$ is lower semicontinuous at every $x \in \mathbb{R}^n$, then $h$ is lower semicontinuous on $\mathbb{R}^n$.

We review a property of convex function, which will be used later.

**Lemma 5** (Mehta et al. 2024)**.** *For a convex function* $h : \mathbb{R}^n \mapsto \bar{\mathbb{R}}$, *if* $x_1 \geqslant x_2$ *and* $y_2 \geqslant y_1$, *then*

$$h(y_1 - x_1) + h(y_2 - x_2) \geqslant h(y_2 - x_1) + h(y_1 - x_2). \tag{26}$$

*Proof.* It can be seen that $(y_2 - x_1)$ and $(y_1 - x_2)$ lie between $(y_1 - x_1)$ and $(y_2 - x_2)$.

$$y_2 - x_1, y_1 - x_2 \in [y_1 - x_1, y_2 - x_2].$$

Hence there exist $\alpha, \beta \in [0, 1]$ such that

$$y_2 - x_1 = \alpha(y_1 - x_1) + (1 - \alpha)(y_2 - x_2),$$
$$y_1 - x_2 = \beta(y_1 - x_1) + (1 - \beta)(y_2 - x_2).$$

Direct calculation indicates that $\alpha = 1 - \beta$. Since $h$ is convex,

$$h(y_2 - x_1) \leqslant \alpha h(y_1 - x_1) + (1 - \alpha)h(y_2 - x_2),$$
$$h(y_1 - x_2) \leqslant (1 - \alpha)h(y_1 - x_1) + \alpha h(y_2 - x_2).$$

Summing up both sides leads to (26). $\qquad\square$

## A.2  DANSKIN'S THEOREM

Suppose the divergence $D(q\|1_n/n)$ is induced by some generating function $\mathfrak{f}$:

$$D(q\|1_n/n) := \frac{1}{n} \sum_{i=1}^{n} \mathfrak{f}(nq_i). \tag{27}$$

Then as long as $\mathfrak{f}$ is $\alpha_n$-strongly convex on $[0, n]$, the mapping $q \mapsto \nu D(q\|1_n/n)$ is ($\alpha_n n\nu$)-strongly convex w.r.t. $\| \cdot \|_2$ on $\mathcal{P}(\sigma)$ (Mehta et al., 2024). Moreover, the mapping $q \mapsto \{\sum_{i=1}^{n} q_i l_i - \nu D(q\|1_n/n)\}$ is strongly concave on $\mathcal{P}(\sigma)$ when $\nu > 0$. Because $\mathcal{P}(\sigma)$ is a compact set, $\{\sum_{i=1}^{n} q_i l_i - \nu D(q\|1_n/n)\}$ has a unique maximizer on $\mathcal{P}(\sigma)$:

$$q^{opt}(l) := \operatorname*{argmax}_{q \in \mathcal{P}(\sigma)} \{q^\top l - \nu D(q\|1_n/n)\}. \tag{28}$$

By Danskin's theorem (Bertsekas, 1999; Mehta et al., 2024), this optimal $q^{opt}(l)$ is exactly $\nabla_l \mathcal{R}_\sigma(l)$. Moreover, $\mathcal{R}_\sigma$ is continuously differentiable with this gradient $\nabla_l \mathcal{R}_\sigma(l)$ as $l \in \mathbb{R}^n$.

### A.3 SOLVING (7) VIA ITS DUAL PROBLEM

We briefly introduce how (7) can be solved via its dual problem (Mehta et al., 2024) for the convenience of reading and understanding. First, we review an important property (Proposition 6.3.1 of (Hiriart-Urruty & Lemaréchal, 2004)) regarding the conjugate of the sum of two closed convex functions $h_1$ and $h_2$:

$$(h_1 + h_2)^*(x) = \inf_{y \in \mathbb{R}^n} \{h_1^*(y) + h_2^*(x - y)\}, \tag{29}$$

where $h^*$ denotes the conjugate function of $h$:

$$h^*(x) = \sup_{y \in \mathbb{R}^n} \{y^\top x - h(y)\}. \tag{30}$$

Furthermore, if $h_1 + h_2$ is strictly convex, we have

$$\nabla(h_1 + h_2)^*(x) = \operatorname*{argmax}_{u \in \mathbb{R}^n} \{u^\top x - (h_1 + h_2)(u)\}, \tag{31}$$

where the maximizer on the right side is unique. If in addition, $h_1^* + h_2^*$ is strictly convex and $h_2^*$ is differentiable, then again by Danskin's theorem (Bertsekas, 1999; Mehta et al., 2024) we have

$$\nabla(h_1 + h_2)^*(x) = \nabla h_2^*(x - y^*(x)) \text{ for } y^*(x) = \operatorname*{argmin}_{y \in \mathbb{R}^n} \{h_1^*(y) + h_2^*(x - y)\}. \tag{32}$$

Next, we look into the specific problem to be solved. Define the indicator function of $\mathcal{P}(\sigma)$ as follows:

$$\iota_{\mathcal{P}(\sigma)}(q) := \begin{cases} 0, & \text{if } q \in \mathcal{P}(\sigma); \\ +\infty, & \text{otherwise.} \end{cases} \tag{33}$$

Then its convex conjugate function is

$$\iota_{\mathcal{P}(\sigma)}^*(l) = \max_{q \in \mathcal{P}(\sigma)} q^\top l. \tag{34}$$

Let $h_1(q) = \iota_{\mathcal{P}(\sigma)}(q)$ and $h_2(q) = \Omega(q) := \nu D(q \| 1_n/n)$. Assume the divergence generating function $\mathfrak{f}$ in (27) and its conjugate $\mathfrak{f}^*$ are both strictly convex, which is satisfied by some common divergences like $\chi^2$-divergence and KL divergence:

$$\mathfrak{f}_{\chi^2}(x) = x^2 - 1, \quad \mathfrak{f}_{\chi^2}^*(y) = \frac{y^2}{4} + 1, \tag{35}$$

$$\mathfrak{f}_{KL}(x) = x \ln x + \iota_+(x), \quad \mathfrak{f}_{KL}^*(y) = \exp(y - 1). \tag{36}$$

Since $D$ in (27) is a sum of $\mathfrak{f}$, it is also strictly convex. Then it follows from (30) that $D^*$ is also strictly convex. Direct calculation yields

$$\sup_{q \in \mathcal{P}(\sigma)} \{q^\top l - \Omega(q)\}$$
$$= \sup_{q \in \mathbb{R}^n} \{q^\top l - (\iota_{\mathcal{P}(\sigma)}(q) + \Omega(q))\}$$
$$= (\iota_{\mathcal{P}(\sigma)} + \Omega)^*(l)$$
$$= \inf_{y \in \mathbb{R}^n} \{\iota_{\mathcal{P}(\sigma)}^*(y) + \Omega^*(l - y)\}$$
$$= \inf_{y \in \mathbb{R}^n} \{\max_{q \in \mathcal{P}(\sigma)} q^\top y + \Omega^*(l - y)\}$$
$$= \inf_{y \in \mathbb{R}^n} \{\sum_{i=1}^n \sigma_i y_{(i)} + \Omega^*(l - y)\}, \tag{37}$$

where $y_{(1)} \leqslant y_{(2)} \leqslant \cdots \leqslant y_{(n)}$ are the sorted components of $y \in \mathbb{R}^n$ in the ascending order. On the other hand, combining (31) and (32) yields

$$\operatorname*{argmax}_{q \in \mathcal{P}(\sigma)} \{q^\top l - \Omega(q)\} = \nabla \Omega^*(l - y^*(l)) \text{ for } y^*(l) = \operatorname*{argmin}_{y \in \mathbb{R}^n} \{\sum_{i=1}^n \sigma_i y_{(i)} + \Omega^*(l - y)\}. \tag{38}$$

(27) indicates that

$$\Omega(q) = \sum_{i=1}^{n} \omega(q_i) \text{ with } \omega(q_i) := \frac{\nu}{n}\mathfrak{f}(nq_i). \tag{39}$$

Direct calculation yields

$$\Omega^*(y) = \sum_{i=1}^{n} \omega^*(y_i) \text{ with } \omega^*(y_i) := \frac{\nu}{n}\mathfrak{f}^*(y_i/\nu). \tag{40}$$

Lemma 5 indicates that if $l_i \leqslant l_j$ and $y_i \geqslant y_j$, then

$$\Omega^*(l_i - y_i) + \Omega^*(l_j - y_j) \geqslant \Omega^*(l_i - y_j) + \Omega^*(l_j - y_i). \tag{41}$$

Hence if $y$ minimizes $\Omega^*(l - y)$ given $l$, the ordering of the components of $y$ must follow that of $l$. That is, for a given permutation $\pi : [n] \mapsto [n]$ such that $l_{\pi(1)} \leqslant \cdots \leqslant l_{\pi(n)}$, we have $y_{\pi(1)} \leqslant \cdots \leqslant y_{\pi(n)}$. On the other hand, the first term of (37) $\sum_{i=1}^{n} \sigma_i y_{(i)}$ does not depend on the ordering of $y$. Then (37) can be rewritten as

$$\inf_{\substack{y \in \mathbb{R}^n \\ y_{\pi(1)} \leqslant \cdots \leqslant y_{\pi(n)}}} \sum_{i=1}^{n} \{\sigma_i y_{\pi(i)} + \frac{\nu}{n}\mathfrak{f}^*(\frac{l_{\pi(i)} - y_{\pi(i)}}{\nu})\}$$

$$= \inf_{\substack{c \in \mathbb{R}^n \\ c_1 \leqslant \cdots \leqslant c_n}} \sum_{i=1}^{n} \{\sigma_i c_i + \frac{\nu}{n}\mathfrak{f}^*(\frac{l_{\pi(i)} - c_i}{\nu})\}$$

$$=: \inf_{\substack{c \in \mathbb{R}^n \\ c_1 \leqslant \cdots \leqslant c_n}} \sum_{i=1}^{n} \mathfrak{g}_i(c_i; l). \tag{42}$$

Once a solution $c^*(l)$ to the right side of (42) is obtained, the corresponding solution to the left side of (42) can be retrieved by $y^*(l) = c^*_{\pi^{-1}}(l)$, where $\pi^{-1}$ denotes the inverse permutation of $\pi$.

Combining (38), (40), and (42), a solution $q^{opt}(l)$ to (7) can be obtained by

$$q_i^{opt}(l) = \frac{1}{n}[\mathfrak{f}^*]'(\frac{l_i - c^{opt}_{\mathrm{rank}(i)}(l)}{\nu}), \quad \forall i \in [n], \tag{43}$$

where $\mathrm{rank}(i)$ denotes the rank of $l_i$ in the sorted form of $l$ in the ascending order. Assuming $l$ is sorted and taking $\mathfrak{f}^*_{\chi^2}$ in (35) as example, we have

$$c^{opt}(l) = \underset{\substack{c \in \mathbb{R}^n \\ c_1 \leqslant \cdots \leqslant c_n}}{\mathrm{argmin}} \sum_{i=1}^{n} \{\sigma_i c_i + \frac{\nu}{4n}(\frac{l_i - c_i}{\nu})^2\}, \tag{44}$$

$$q_i^{opt}(l) = \frac{1}{2n\nu}(l_i - c_i^{opt}(l)). \tag{45}$$

(44) can be solved by letting the derivative w.r.t. each $c_i$ being zero, which yields $c_i^{opt}(l) = l_i + 2n\nu\sigma_i$. Then $q_i^{opt}(l) = \sigma_i$.

## A.4 Lipschitz Continuity of $q^{opt}(l)$ and $q^{opt}(\ell(w))$

**Lemma 6** (Mehta et al. 2024). *Let $\ell : \mathbb{R}^d \mapsto \mathbb{R}^n$ be differentiable with Jacobian $\nabla\ell(w) \in \mathbb{R}^{n \times d}$ and each $\ell_i$ be $\mu$-strongly convex. Let $\nu > 0$ and $\mathfrak{f}$ be $\alpha_n$-strongly convex on $[0, n]$. Then $f$ defined in (3) is differentiable with its gradient being*

$$\nabla f(w) = (\nabla\ell(w))^\top q^{opt}(\ell(w)). \tag{46}$$

*Moreover, $l \mapsto q^{opt}(l)$ is $L_q := (\alpha_n n\nu)^{-1}$-Lipschitz continuous.*

*Proof.* From Appendix A.2, $q \mapsto \nu D(q\|1_n/n)$ is $(\alpha_n n\nu)$-strongly convex w.r.t. $\|\cdot\|_2$ on $[0,1]^n$. Because each $\ell_i$ is convex w.r.t. $w$, any $q \in \mathcal{P}(\sigma)$ is nonnegative, and the mapping

$$w \mapsto \max_{q \in \mathcal{P}(\sigma)} \{q^\top \ell(w) - \nu D(q\|1_n/n)\} \tag{47}$$

is a $w$-point-wise maximum of $\{q^\top \ell(w) - \nu D(q\|1_n/n)\}$ over $q$, this mapping is convex w.r.t. $w$. Then by Danskin's theorem in Appendix A.2, $f$ is differentiable with

$$\nabla f(w) = (\nabla \ell(w))^\top q^{opt}(\ell(w)).$$

Moreover, Theorem 1 of (Nesterov, 2005) implies that $l \mapsto q^{opt}(l)$ is Lipschitz continuous with Lipschitz constant being the inverse of the strong convexity constant of $\nu D(q\|1_n/n)$, which is $L_q := (\alpha_n n\nu)^{-1}$. $\qquad\square$

**Lemma 7** (Mehta et al. 2024). *For any* $w, w' \in \mathbb{R}^d$,

$$\|q^{opt}(\ell(w)) - q^{opt}(\ell(w'))\|_2^2 \leqslant \frac{G^2}{n\alpha_n^2\nu^2}\|w - w'\|_2^2 =: L_{q\circ\ell}^2\|w - w'\|_2^2. \tag{48}$$

*Thus the mapping* $w \mapsto q^{opt}(\ell(w))$ *is* $L_{q\circ\ell}$-*Lipschitz continuous.*

*Proof.*

$$\|q^{opt}(\ell(w)) - q^{opt}(\ell(w'))\|_2^2$$

$$\leqslant \frac{1}{n^2\alpha_n^2\nu^2}\|\ell(w) - \ell(w')\|_2^2 \tag{49}$$

$$= \frac{1}{n^2\alpha_n^2\nu^2}\sum_{i=1}^n (\ell_i(w) - \ell_i(w'))^2$$

$$\leqslant \frac{1}{n^2\alpha_n^2\nu^2}\cdot n \cdot G^2\|w - w'\|_2^2 \tag{50}$$

$$= \frac{G^2}{n\alpha_n^2\nu^2}\|w - w'\|_2^2$$

$$=: L_{q\circ\ell}^2\|w - w'\|_2^2. \tag{51}$$

The inequality of (49) is from Lemma 6, and the inequality of (50) is from the $G$-Lipschitz continuity of $\ell_i$. $\qquad\square$

## A.5 Proof of Lemma 1

*Proof.* We have $\|q^{opt}(\ell(w))\|_2 \leqslant M_q$ for any $w \in \mathbb{R}^d$ since $q$ lies in a compact set $\mathcal{P}(\sigma)$.

$$\|\nabla f(w) - \nabla f(w')\|_2$$
$$= \|(\nabla \ell(w))^\top q^{opt}(\ell(w)) - (\nabla \ell(w'))^\top q^{opt}(\ell(w'))\|_2$$
$$= \|(\nabla \ell(w))^\top q^{opt}(\ell(w)) - (\nabla \ell(w'))^\top q^{opt}(\ell(w)) + (\nabla \ell(w'))^\top q^{opt}(\ell(w)) - (\nabla \ell(w'))^\top q^{opt}(\ell(w'))\|_2$$
$$\leqslant \|q^{opt}(\ell(w))\|_2\|\nabla \ell(w) - \nabla \ell(w')\|_2 + \|\nabla \ell(w')\|_2\|q^{opt}(\ell(w)) - q^{opt}(\ell(w'))\|_2$$
$$\leqslant M_q L_\ell\|w - w'\|_2 + \|\nabla \ell(w')\|_2 L_{q\circ\ell}\|w - w'\|_2$$
$$= (M_q L_\ell + \|\nabla \ell(w')\|_2 L_{q\circ\ell})\|w - w'\|_2$$
$$\leqslant (M_q L_\ell + \sqrt{n}G L_{q\circ\ell})\|w - w'\|_2$$
$$=: L_f\|w - w'\|_2. \tag{52}$$

It indicates that $f$ is Lipschitz smooth, or $L_f$-smooth to specify the constant $L_f$. $\qquad\square$

### A.6 PROOF OF LEMMA 2

*Proof.* First, we verify that $\Delta^{(t)}$ has zero mean conditioned on time $t$.

$$
\begin{aligned}
\mathbb{E}_t[\Delta^{(t)}] &= \mathbb{E}_t[v^{(t)} - \nabla f(w^{(t)})] \\
&= \mathbb{E}_t[nq_{i_t}^{(t)}\nabla\ell_{i_t}(w^{(t)}) - (n\rho_{i_t}^{(t)}\nabla\ell_{i_t}(z_{i_t}^{(t)}) - \sum_{i=1}^{n}\rho_i^{(t)}\nabla\ell_i(z_i^{(t)})) - \sum_{i=1}^{n}q_i^{(t)}\nabla\ell_i(w^{(t)})] \\
&= \mathbb{E}_t[nq_{i_t}^{(t)}\nabla\ell_{i_t}(w^{(t)}) - n\rho_{i_t}^{(t)}\nabla\ell_{i_t}(z_{i_t}^{(t)})] + \sum_{i=1}^{n}\rho_i^{(t)}\nabla\ell_i(z_i^{(t)}) - \sum_{i=1}^{n}q_i^{(t)}\nabla\ell_i(w^{(t)}) \\
&= \sum_{i=1}^{n}q_i^{(t)}\nabla\ell_i(w^{(t)}) - \sum_{i=1}^{n}\rho_i^{(t)}\nabla\ell_i(z_i^{(t)}) + \sum_{i=1}^{n}\rho_i^{(t)}\nabla\ell_i(z_i^{(t)}) - \sum_{i=1}^{n}q_i^{(t)}\nabla\ell_i(w^{(t)}) \\
&= 0.
\end{aligned}
\tag{53}
$$

Next, we turn to bound the variance of $\Delta^{(t)}$ conditioned on time $t$.

$$
\mathbb{E}_t[\|\Delta^{(t)}\|_2^2] = \mathbb{E}_t[\|v^{(t)} - \nabla f(w^{(t)})\|_2^2]
$$

$$
= \mathbb{E}_t[\|(nq_{i_t}^{(t)}\nabla\ell_{i_t}(w^{(t)}) - \sum_{i=1}^{n}q_i^{(t)}\nabla\ell_i(w^{(t)})) - (n\rho_{i_t}^{(t)}\nabla\ell_{i_t}(z_{i_t}^{(t)}) - \sum_{i=1}^{n}\rho_i^{(t)}\nabla\ell_i(z_i^{(t)}))\|_2^2]
$$

$$
= \mathbb{E}_t[\|(nq_{i_t}^{(t)}\nabla\ell_{i_t}(w^{(t)}) - n\rho_{i_t}^{(t)}\nabla\ell_{i_t}(z_{i_t}^{(t)})) - (\sum_{i=1}^{n}q_i^{(t)}\nabla\ell_i(w^{(t)}) - \sum_{i=1}^{n}\rho_i^{(t)}\nabla\ell_i(z_i^{(t)}))\|_2^2]
\tag{54}
$$

$$
= \mathbb{E}_t[\|nq_{i_t}^{(t)}\nabla\ell_{i_t}(w^{(t)}) - n\rho_{i_t}^{(t)}\nabla\ell_{i_t}(z_{i_t}^{(t)})\|_2^2] - \|\sum_{i=1}^{n}q_i^{(t)}\nabla\ell_i(w^{(t)}) - \sum_{i=1}^{n}\rho_i^{(t)}\nabla\ell_i(z_i^{(t)})\|_2^2
\tag{55}
$$

$$
\leqslant \mathbb{E}_t[\|nq_{i_t}^{(t)}\nabla\ell_{i_t}(w^{(t)}) - n\rho_{i_t}^{(t)}\nabla\ell_{i_t}(z_{i_t}^{(t)})\|_2^2]
$$

$$
= \mathbb{E}_t[\|(nq_{i_t}^{(t)}\nabla\ell_{i_t}(w^{(t)}) - nq_{i_t}^{*}\nabla\ell_{i_t}(w^{*})) - (n\rho_{i_t}^{(t)}\nabla\ell_{i_t}(z_{i_t}^{(t)}) - nq_{i_t}^{*}\nabla\ell_{i_t}(w^{*}))\|_2^2]
$$

$$
\leqslant 2\mathbb{E}_t[\|nq_{i_t}^{(t)}\nabla\ell_{i_t}(w^{(t)}) - nq_{i_t}^{*}\nabla\ell_{i_t}(w^{*})\|_2^2] + 2\mathbb{E}_t[\|n\rho_{i_t}^{(t)}\nabla\ell_{i_t}(z_{i_t}^{(t)}) - nq_{i_t}^{*}\nabla\ell_{i_t}(w^{*})\|_2^2]
\tag{56}
$$

$$
= \frac{2}{n}\sum_{i=1}^{n}\|nq_i^{(t)}\nabla\ell_i(w^{(t)}) - nq_i^{*}\nabla\ell_i(w^{*})\|_2^2 + \frac{2}{n}\sum_{i=1}^{n}\|n\rho_i^{(t)}\nabla\ell_i(z_i^{(t)}) - nq_i^{*}\nabla\ell_i(w^{*})\|_2^2
$$

$$
= 2Q^{(t)} + 2S^{(t)}.
\tag{57}
$$

In (54), we have

$$
\mathbb{E}_t[nq_{i_t}^{(t)}\nabla\ell_{i_t}(w^{(t)}) - n\rho_{i_t}^{(t)}\nabla\ell_{i_t}(z_{i_t}^{(t)})] = \sum_{i=1}^{n}q_i^{(t)}\nabla\ell_i(w^{(t)}) - \sum_{i=1}^{n}\rho_i^{(t)}\nabla\ell_i(z_i^{(t)}).
\tag{58}
$$

Thus (54) is actually the variance $\text{Var}[nq_{i_t}^{(t)}\nabla\ell_{i_t}(w^{(t)}) - n\rho_{i_t}^{(t)}\nabla\ell_{i_t}(z_{i_t}^{(t)})]$. Then (55) can be derived from (54) by using a property of the variance: $\text{Var}[\|a\|_2^2] = \mathbb{E}[\|a\|_2^2] - \|\mathbb{E}[a]\|_2^2$. The inequality (56) is an arithmetic mean-geometric mean (AM-GM) inequality.

As for the split of $\mathbb{E}_t[S^{(t+1)}]$,

$$
\mathbb{E}_t[S^{(t+1)}]
$$

$$
= \frac{1}{n}\sum_{i=1}^{n}\mathbb{E}_t[\|n\rho_{i_{t+1}}^{(t+1)}\nabla\ell_{i_{t+1}}(z_{i_{t+1}}^{(t+1)}) - nq_{i_{t+1}}^{*}\nabla\ell_{i_{t+1}}(w^{*})\|_2^2]
$$

$$
= \frac{1}{n}\sum_{i=1}^{n}[\frac{1}{n}\|nq_i^{(t)}\nabla\ell_i(w^{(t)}) - nq_i^{*}\nabla\ell_i(w^{*})\|_2^2 + (1 - \frac{1}{n})\|n\rho_i^{(t)}\nabla\ell_i(z_i^{(t)}) - nq_i^{*}\nabla\ell_i(w^{*})\|_2^2]
$$

$$
= \frac{1}{n}Q^{(t)} + (1 - \frac{1}{n})S^{(t)}.
\tag{59}
$$

The conditional expectation of $\rho_{i_{t+1}}^{(t+1)}$ (or $z_{i_{t+1}}^{(t+1)}$) on time $t$ is calculated by $q_i^{(t)}$ (or $w^{(t)}$) with probability $\frac{1}{n}$ plus $\rho_i^{(t)}$ (or $z_i^{(t)}$) with probability $(1 - \frac{1}{n})$. Then the second equality holds from the law of the unconscious statistician (LOTUS). $\qquad\square$

### A.7 PROOF OF LEMMA 3

*Proof.* First, we recall the definition of proximal mapping

$$w^{(t+1)} = \text{prox}_{\eta g}(w^{(t)} - \eta v^{(t)}) = \underset{w \in \mathbb{R}^d}{\text{argmin}}\{\frac{1}{2}\|w - (w^{(t)} - \eta v^{(t)})\|_2^2 + \eta g(w)\}. \qquad (60)$$

Since $\eta g(w)$ is convex, Fermat's rule (Proposition 6.a in Moreau 1965) indicates that $w^{(t+1)}$ as a minimizer of (60) satisfies

$$-w^{(t+1)} + (w^{(t)} - \eta v^{(t)}) \in \partial(\eta g(w^{(t+1)})) = \eta \partial g(w^{(t+1)}), \qquad (61)$$

where $\partial g(w^{(t+1)})$ denotes the subdifferential of $g$ at $w^{(t+1)}$:

$$\partial g(w^{(t+1)}) := \{y \in \mathbb{R}^d : g(w) \geqslant g(w^{(t+1)}) + \langle y, w - w^{(t+1)}\rangle, \forall w \in \mathbb{R}^d\}. \qquad (62)$$

Inserting $y = (w^{(t)} - w^{(t+1)})/\eta - v^{(t)} = h^{(t)} - v^{(t)}$ into (62) and using $y \in \partial g(w^{(t+1)})$ from (61), we have

$$g(w) \geqslant g(w^{(t+1)}) + \langle h^{(t)} - v^{(t)}, w - w^{(t+1)}\rangle, \forall w \in \mathbb{R}^d. \qquad (63)$$

On the other hand, since $f(w)$ is $\mu$-strongly convex and $L_f$-smooth, we have

$$f(w) \geqslant f(w^{(t)}) + \langle \nabla f(w^{(t)}), w - w^{(t)}\rangle + \frac{\mu}{2}\|w - w^{(t)}\|_2^2, \qquad (64)$$

$$f(w^{(t)}) \geqslant f(w^{(t+1)}) - \langle \nabla f(w^{(t)}), w^{(t+1)} - w^{(t)}\rangle - \frac{L_f}{2}\|w^{(t+1)} - w^{(t)}\|_2^2. \qquad (65)$$

(64) can be deduced from the first-derivative condition of a convex function, while (65) is an important property of $L_f$-smoothness indicated by Proposition A.24 of (Bertsekas, 1999). Summing up both sides of (63), (64), and (65), and noticing that $f(w) + g(w) = F(w)$ and $f(w^{(t+1)}) + g(w^{(t+1)}) = F(w^{(t+1)})$, we have

$$\begin{aligned} F(w) \geqslant &F(w^{(t+1)}) + \langle \nabla f(w^{(t)}), w - w^{(t+1)}\rangle - \frac{L_f}{2}\|w^{(t+1)} - w^{(t)}\|_2^2 \\ &+ \frac{\mu}{2}\|w - w^{(t)}\|_2^2 + \langle h^{(t)} - v^{(t)}, w - w^{(t+1)}\rangle \\ =&F(w^{(t+1)}) - \frac{L_f}{2}\|w^{(t+1)} - w^{(t)}\|_2^2 + \frac{\mu}{2}\|w - w^{(t)}\|_2^2 \\ &+ \langle \Delta^{(t)}, w^{(t+1)} - w\rangle + \langle h^{(t)}, w - w^{(t)}\rangle + \langle h^{(t)}, w^{(t)} - w^{(t+1)}\rangle. \end{aligned} \qquad (66)$$

Inserting $w^{(t)} - w^{(t+1)} = \eta h^{(t)}$ into (66) yields

$$\begin{aligned} F(w) \geqslant &F(w^{(t+1)}) + (1 - \frac{L_f\eta}{2})\eta\|h^{(t)}\|_2^2 + \frac{\mu}{2}\|w - w^{(t)}\|_2^2 \\ &+ \langle \Delta^{(t)}, w^{(t+1)} - w\rangle + \langle h^{(t)}, w - w^{(t)}\rangle. \end{aligned} \qquad (67)$$

Since $\eta \leqslant \frac{1}{L_f}$, (67) implies

$$F(w) \geqslant F(w^{(t+1)}) + \frac{\eta}{2}\|h^{(t)}\|_2^2 + \frac{\mu}{2}\|w - w^{(t)}\|_2^2 + \langle \Delta^{(t)}, w^{(t+1)} - w\rangle + \langle h^{(t)}, w - w^{(t)}\rangle. \qquad (68)$$

$\qquad\square$

## A.8 PROOF OF LEMMA 4

*Proof.* First, we represent the optimality gap of $w^{(t+1)}$ by that of $w^{(t)}$:

$$
\begin{aligned}
\|w^{(t+1)} - w^*\|_2^2 &= \|w^{(t)} - \eta h^{(t)} - w^*\|_2^2 \\
&= \|w^{(t)} - w^*\|_2^2 - 2\eta\langle h^{(t)}, w^{(t)} - w^*\rangle + \eta^2\|h^{(t)}\|_2^2.
\end{aligned}
\tag{69}
$$

Then we can exploit the descent property (20) in Lemma 3 to suppress the optimality gap:

$$
-\langle h^{(t)}, w^{(t)} - w^*\rangle + \frac{\eta}{2}\|h^{(t)}\|_2^2 \leqslant F(w^*) - F(w^{(t+1)}) - \frac{\mu}{2}\|w^* - w^{(t)}\|_2^2 - \langle \Delta^{(t)}, w^{(t+1)} - w^*\rangle.
\tag{70}
$$

Connecting the left side of (70) to the right side of (69) yields:

$$
\|w^{(t+1)} - w^*\|_2^2 \leqslant (1 - \frac{\mu}{2})\|w^{(t)} - w^*\|_2^2 + 2\eta[F(w^*) - F(w^{(t+1)})] - 2\eta\langle \Delta^{(t)}, w^{(t+1)} - w^*\rangle.
\tag{71}
$$

Since the full proximal gradient step

$$
\bar{w}^{(t+1)} = \text{prox}_{\eta g}(w^{(t)} - \eta \sum_{i=1}^n \rho_i^{(t)} \nabla \ell_i(z_i^{(t)}))
\tag{72}
$$

is independent of $i_t$, it can be exploited to separate the stochastic terms from the deterministic terms. Specifically,

$$
\begin{aligned}
&- 2\eta\langle \Delta^{(t)}, w^{(t+1)} - w^*\rangle \\
=&- 2\eta\langle \Delta^{(t)}, w^{(t+1)} - \bar{w}^{(t+1)}\rangle - 2\eta\langle \Delta^{(t)}, \bar{w}^{(t+1)} - w^*\rangle \\
\leqslant&2\eta\|\Delta^{(t)}\|_2\|w^{(t+1)} - \bar{w}^{(t+1)}\|_2 - 2\eta\langle \Delta^{(t)}, \bar{w}^{(t+1)} - w^*\rangle \\
\leqslant&2\eta\|\Delta^{(t)}\|_2\|(w^{(t)} - \eta v^{(t)}) - (w^{(t)} - \eta\nabla f(w^{(t)}))\|_2 - 2\eta\langle \Delta^{(t)}, \bar{w}^{(t+1)} - w^*\rangle \\
=&2\eta^2\|\Delta^{(t)}\|_2^2 - 2\eta\langle \Delta^{(t)}, \bar{w}^{(t+1)} - w^*\rangle.
\end{aligned}
\tag{73}
$$

Connecting the left side of (73) to the right side of (71) yields

$$
\begin{aligned}
\|w^{(t+1)} - w^*\|_2^2 \leqslant&(1 - \frac{\mu}{2})\|w^{(t)} - w^*\|_2^2 - 2\eta[F(w^{(t+1)}) - F(w^*)] \\
&+ 2\eta^2\|\Delta^{(t)}\|_2^2 - 2\eta\langle \Delta^{(t)}, \bar{w}^{(t+1)} - w^*\rangle.
\end{aligned}
\tag{74}
$$

Taking expectation on both sides of (74) conditioned on $i_t$ yields

$$
\begin{aligned}
\mathbb{E}_t[\|w^{(t+1)} - w^*\|_2^2] \leqslant&(1 - \frac{\mu}{2})\|w^{(t)} - w^*\|_2^2 - 2\eta[\mathbb{E}_t[F(w^{(t+1)})] - F(w^*)] \\
&+ 2\eta^2\mathbb{E}_t[\|\Delta^{(t)}\|_2^2] - 2\eta\mathbb{E}_t[\langle \Delta^{(t)}, \bar{w}^{(t+1)} - w^*\rangle].
\end{aligned}
\tag{75}
$$

Since $\bar{w}^{(t+1)}$ and $w^*$ are independent of $i_t$, it follows from (16) that

$$
\mathbb{E}_t[\langle \Delta^{(t)}, \bar{w}^{(t+1)} - w^*\rangle] = \langle \mathbb{E}_t[\Delta^{(t)}], \bar{w}^{(t+1)} - w^*\rangle = 0.
\tag{76}
$$

By this means, the first-order stochastic term of $\Delta^{(t)}$ can be eliminated. Inserting (17) and (76) into (75) yields

$$
\mathbb{E}_t[\|w^{(t+1)} - w^*\|_2^2] \leqslant (1 - \frac{\mu}{2})\|w^{(t)} - w^*\|_2^2 - 2\eta[\mathbb{E}_t[F(w^{(t+1)})] - F(w^*)] + 4\eta^2(Q^{(t)} + S^{(t)}).
\tag{77}
$$

$\square$

### A.9 PROOF OF THEOREM 1

*Proof.* First, we verify the linear convergence of the Lyapunov function sequence

$$\mathbb{E}_t[V^{(t+1)}] \leqslant (1-\tau)V^{(t)}. \tag{78}$$

To do this, we need to bound the error term:

$$
\begin{aligned}
Q^{(t)} :=& \frac{1}{n}\sum_{i=1}^{n} \|nq_i^{(t)}\nabla\ell_i(w^{(t)}) - nq_i^*\nabla\ell_i(w^*)\|_2^2 \\
=& n\|\nabla f(w^{(t)}) - \nabla f(w^*)\|_2^2 \\
\leqslant& nL_f\|w^{(t)} - w^*\|_2^2. 
\end{aligned} \tag{79}
$$

Then by direct calculations,

$$
\begin{aligned}
\mathbb{E}_t[V^{(t+1)}] &= \mathbb{E}_t[\|w^{(t+1)} - w^*\|_2^2] + c_1 \cdot 4\eta^2\mathbb{E}_t[S^{(t+1)}] \\
&\leqslant (1-\frac{\mu}{2})\|w^{(t)} - w^*\|_2^2 - 2\eta(\mathbb{E}_t[F(w^{(t+1)})] - F(w^*)) + 4\eta^2(Q^{(t)} + S^{(t)}) + c_1 \cdot 4\eta^2\mathbb{E}_t[S^{(t+1)}] \\
&\leqslant (1-\frac{\mu}{2})\|w^{(t)} - w^*\|_2^2 + 4\eta^2(Q^{(t)} + S^{(t)}) + c_1 \cdot 4\eta^2\mathbb{E}_t[S^{(t+1)}] \\
&\leqslant (1-\frac{\mu}{2})\|w^{(t)} - w^*\|_2^2 + 4\eta^2(1+\frac{c_1}{n})Q^{(t)} + 4\eta^2(1+c_1-\frac{c_1}{n})S^{(t)} \\
&\leqslant (1-\frac{\mu}{2})\|w^{(t)} - w^*\|_2^2 + 4\eta^2 L_f(n+c_1)\|w^{(t)} - w^*\|_2^2 + 4\eta^2(1+c_1-\frac{c_1}{n})S^{(t)} \\
&\leqslant (1-\frac{\mu}{2} + 4\eta^2 L_f(n+c_1))\|w^{(t)} - w^*\|_2^2 + 4\eta^2(1+c_1-\frac{c_1}{n})S^{(t)}. 
\end{aligned} \tag{80}
$$

We can properly set the learning step size $\eta \in (0, \sqrt{\frac{\mu}{8L_f(n+c_1)}})$ and the coefficient $c_1 > n$, then a suitable $\tau$ can be found to satisfy (78):

$$\tau \in (0, \min\{\frac{\mu}{2} - 4\eta^2 L_f(n+c_1), \frac{1}{n} - \frac{1}{c_1}\}]. \tag{81}$$

Second, we verify the linear convergence of the iterate sequence $\{w^{(t)}\}_{t\geqslant 1}$. To do this, we need to bound the initial term:

$$
\begin{aligned}
S^{(0)} :=& \frac{1}{n}\sum_{i=1}^{n} \|n\rho_i^{(0)}\nabla\ell_i(z_i^{(0)}) - nq_i^*\nabla\ell_i(w^*)\|_2^2 \\
=& \frac{1}{n}\sum_{i=1}^{n} \|nq_i^{(0)}\nabla\ell_i(w^{(0)}) - nq_i^*\nabla\ell_i(w^*)\|_2^2 \\
=& \frac{1}{n}\sum_{i=1}^{n} \|(nq_i^{(0)}\nabla\ell_i(w^{(0)}) - nq_i^{(0)}\nabla\ell_i(w^*)) + (nq_i^{(0)}\nabla\ell_i(w^*) - nq_i^*\nabla\ell_i(w^*))\|_2^2 \\
\leqslant& \frac{2}{n}\sum_{i=1}^{n} \|nq_i^{(0)}(\nabla\ell_i(w^{(0)}) - \nabla\ell_i(w^*))\|_2^2 + \frac{2}{n}\sum_{i=1}^{n} \|n(q_i^{(0)} - q_i^*)\nabla\ell_i(w^*)\|_2^2 \\
\leqslant& 2n\sum_{i=1}^{n}(q_i^{(0)})^2 L_\ell^2\|w^{(0)} - w^*\|_2^2 + 2nG^2\sum_{i=1}^{n}\|q_i^{(0)} - q_i^*\|_2^2 \\
=& 2n\|q^{(0)}\|_2^2 L_\ell^2\|w^{(0)} - w^*\|_2^2 + 2nG^2\|q^{(0)} - q^*\|_2^2 \\
\leqslant& 2n\|q^{(0)}\|_2^2 L_\ell^2\|w^{(0)} - w^*\|_2^2 + 2nG^2 L_{q\circ\ell}^2\|w^{(0)} - w^*\|_2^2 \\
=& 2n(\|q^{(0)}\|_2^2 L_\ell^2 + G^2 L_{q\circ\ell}^2)\|w^{(0)} - w^*\|_2^2. 
\end{aligned} \tag{82}
$$

On the other hand, since $c_1, \eta$, and $S^{(t)} \geqslant 0$, $\|w^{(t)} - w^*\|_2^2 \leqslant V^{(t)}$. Using (78) recursively, we have

$$\mathbb{E}[\|w^{(t)} - w^*\|_2^2]$$

$$=\mathbb{E}_0[\|w^{(t)} - w^*\|_2^2]$$
$$\leqslant \mathbb{E}_0[V^{(t)}] = \mathbb{E}_0[\mathbb{E}_{t-1}[V^{(t)}]]$$
$$\leqslant (1-\tau)\mathbb{E}_0[V^{(t-1)}] \leqslant \cdots$$
$$\leqslant (1-\tau)^t V^{(0)}$$
$$=(1-\tau)^t(\|w^{(0)} - w^*\|_2^2 + c_1 \cdot 4\eta^2 S^{(0)})$$
$$\leqslant (1-\tau)^t(1 + 8c_1 n\eta^2(\|q^{(0)}\|_2^2 L_\ell^2 + G^2 L_{q\circ\ell}^2))\|w^{(0)} - w^*\|_2^2. \tag{83}$$

We can set some specific values for the coefficients and hyperparameters that satisfy linear convergence. For example,

$$c_1 = 2n, \quad \eta = \frac{1}{2}\sqrt{\frac{\mu}{8L_f(n+c_1)}}. \tag{84}$$

Then as long as $n$ is sufficiently large, we can set $\tau = \frac{1}{3n}$. From (51) and (52), we have

$$L_{q\circ\ell}^2 = \frac{G^2}{n\alpha_n^2\nu^2}, \quad L_f = M_q L_\ell + \sqrt{n}G L_{q\circ\ell}. \tag{85}$$

Inserting (84), (85), and $\tau = \frac{1}{3n}$ into (83) yields

$$\mathbb{E}[\|w^{(t)} - w^*\|_2^2]$$
$$\leqslant (1 - \frac{1}{3n})^t(1 + \frac{\mu n}{6(M_q L_\ell + \frac{G^2}{\alpha_n\nu})}(\|q^{(0)}\|_2^2 L_\ell^2 + \frac{G^4}{n\alpha_n^2\nu^2}))\|w^{(0)} - w^*\|_2^2$$
$$=(1 - \frac{1}{3n})^t(1 + \frac{\mu G^4}{6(M_q L_\ell \alpha_n\nu + G^2)\alpha_n\nu} + \frac{\|q^{(0)}\|_2^2 L_\ell^2 \mu n}{6(M_q L_\ell + \frac{G^2}{\alpha_n\nu})})\|w^{(0)} - w^*\|_2^2$$
$$=:(1 - \frac{1}{3n})^t(C_3 + C_1\mu n)C_2, \tag{86}$$

where $C_1, C_2, C_3 \geqslant 0$ are three constants independent of $n$.

In order to achieve $\mathbb{E}[\|w^{(t)} - w^*\|_2^2] \leqslant \varepsilon$ for a sufficiently small $\varepsilon > 0$, the number of iterations required for PSG-SRM is

$$t = \mathcal{O}\left(\ln\left(\frac{(C_3 + C_1\mu n)C_2}{\varepsilon}\right) / \ln\left(\frac{3n}{3n-1}\right)\right), \tag{87}$$

Let $(1 - \frac{1}{3n})^t \leqslant \exp(-\frac{t}{3n})$ in (86), then (87) becomes

$$t = \mathcal{O}\left(n \cdot \ln\left(\frac{(C_3 + C_1\mu n)C_2}{\varepsilon}\right)\right), \tag{88}$$

which is a lower computational complexity than that of PROSPECT (Mehta et al., 2024) considering both $n$ and $\varepsilon$. The number of iterations for PROSPECT is

$$t = \mathcal{O}\left(n \cdot \ln\left(\frac{(C_3' + C_4'n^2 + C_1'n)C_2'}{\varepsilon}\right)\right), \quad C_1', C_2', C_3', C_4' \geqslant 0, \tag{89}$$

which has an extra higher-order term $C_4'n^2$ in the nominator inside the $\ln$ function compared with (88). The reason is that the Lyapunov function of PROSPECT uses $c_1 \cdot S^{(0)}$ instead of $c_1 \cdot \eta^2 S^{(0)}$ and its learning rate $\eta$ is not divided by any factor of $n$. This computational complexity may not be further reduced based on the current algorithmic framework of PROSPECT. □

# B IMPLEMENTATION DETAILS AND ADDITIONAL RESULTS FOR EXPERIMENTS

## B.1 SPECTRAL RISK MEASURES

Given a spectrum $\sigma := (\sigma_1, \cdots, \sigma_n)$ with $\sigma_1 \leqslant \cdots \leqslant \sigma_n$, the uncertainty set is defined as

$$\mathcal{P}(\sigma) := \text{ConvexHull}\{\text{permutations of } \sigma\}$$

and the corresponding spectral risk measure is

$$\min_{w \in \mathbb{R}^d} \mathcal{R}_{\mathcal{P}(\sigma)}(\ell(w)) \text{ for } \mathcal{R}_{\mathcal{P}(\sigma)}(l) := \max_{q \in \mathcal{P}(\sigma)} \left\{ \sum_{i=1}^{n} q_i l_i - \nu D(q \| 1_n/n) \right\},$$

where $\ell(w) = (\ell_1(w), \cdots, \ell_n(w)) \in \mathbb{R}^n$ with $\ell_i(w) := \ell(w, x_i, y_i)$ being the loss of the $i$-th training sample for supervised learning.

We consider three types of spectral risk measures: $p$-CVaR, $b$-extremile, and $\varsigma$-ESRM, defined as follows:

$$\sigma_i = \left\{ \begin{array}{ll} \frac{1}{np} & \text{if } i \in \{\lceil n(1-p) \rceil, \cdots, n\} \\ 1 - \frac{\lfloor np \rfloor}{np} & \text{if } \lfloor n(1-p) \rfloor < i < \lceil n(1-p) \rceil \end{array} \right. , \quad p \in (0,1); \qquad (p\text{-CVaR})$$

$$\sigma_i = \left( \frac{i}{n} \right)^b - \left( \frac{i-1}{n} \right)^b, \quad b \geq 1; \qquad (b\text{-extremile})$$

$$\sigma_i = \frac{e^\varsigma \left( e^{\varsigma i/n} - e^{\varsigma(i-1)/n} \right)}{1 - e^{-\varsigma}}, \quad \varsigma > 0. \qquad (\varsigma\text{-ESRM})$$

In our experiments, we set $p = 0.5$, $b = 2$, and $\varsigma = 1.0$ for CVaR, extremile, and ESRM, respectively.

## B.2 Loss Functions

For regression tasks, given a training sample $(x_i, y_i) \in \mathbb{R}^d \times \mathbb{R}$, we use the squared loss

$$\ell_i(w) := \frac{1}{2} \left( y_i - w^\top x_i \right)^2.$$

For multi-class classification tasks, given a training sample $(x_i, y_i) \in \mathbb{R}^d \times \{1, 2, \ldots, C\}$, we use the multinomial logistic loss

$$\ell_i(w) := -\log p((x_i, y_i); w), \quad \text{where } p((x_i, y_i); w) := \frac{\exp(w_{y_i}^T x_i)}{\sum\limits_{j=1}^{C} \exp(w_j^T x_i)}.$$

In regression tasks, the parameter $w$ is a $d$-dimensional column vector. As for multi-class classification tasks, $w$ is a $d \times C$ matrix, where $w_j$ denotes the $j$-th column of $w$.

## B.3 Data Sets

We provide profiles of the data sets used for the experiments in Table A1. For more details about these data sets, please refer to (Mehta et al., 2024).

Table A1: Profiles of data sets.

| Data Set | $d$ | $n_{train}$ | $n_{val}$ | $n_{test}$ | Task |
|---|---|---|---|---|---|
| yacht | 6 | 244 | 31 | 31 | Regression |
| energy | 8 | 614 | 77 | 77 | Regression |
| concrete | 8 | 824 | 103 | 103 | Regression |
| kin8nm | 8 | 6553 | 819 | 820 | Regression |
| power | 4 | 7654 | 957 | 957 | Regression |
| acsincome | 202 | 4000 | 500 | 500 | Regression |
| amazon | 137 | 10000 | 5000 | 5000 | Multi-class Classification |
| iwildcam | 157 | 20000 | 5000 | 5000 | Multi-class Classification |

### B.4 HYPERPARAMETERS FOR PSG-SRM

The regularization parameter $\lambda$ is selected from the set $\{0.01, 0.1, 1, 10, 100\}$, the learning rate $\eta$ is selected from the set $\{1e-5, 3e-5, 1e-4, 3e-4, 1e-3, 3e-3, 0.01, 0.03, 0.1\}$, and the shift cost $\nu$ is selected from the set $\{0.01, 0.1, 1, 10, 100\}$. The random seed $k$ that determines algorithmic randomness is drawn from $\{1, 2, \cdots, 10\}$. We perform a grid search to determine the optimal $\lambda$, $\eta$, and $\nu$, then evaluate parameter configurations based on the average validation loss across different random seeds. Specifically, we define the validation loss as

$$\mathcal{L}_k(\lambda, \eta, \nu) = \frac{1}{n_{val}} \sum_{i=1}^{n_{val}} \ell_i(w_k^*(\lambda, \eta, \nu)),$$

where $w_k^*(\lambda, \eta, \nu)$ is learned on the training set by using specific values of $\lambda$, $\eta$, $\nu$, and $k$. The final values of $\lambda$, $\eta$, and $\nu$ are chosen such that the average validation loss $\frac{1}{10} \sum_{k=1}^{10} \mathcal{L}_k(\lambda, \eta, \nu)$ is minimized.

### B.5 PROXIMAL OPERATOR FOR SOREL

The implementation of SOREL involves computing the following proximal operator:

$$\text{prox}_{\alpha(g + \frac{1}{2\tau} \|\cdot - u\|^2)}(v) := \underset{w}{\text{argmin}} \left\{ \alpha g(w) + \frac{\alpha}{2\tau} \|w - u\|^2 + \frac{1}{2} \|w - v\|^2 \right\}, \quad \alpha \geqslant 0, \tau > 0,$$

which can be simplified as

$$\text{prox}_{\frac{\tau\alpha}{\alpha + \tau} g} \left( \frac{\alpha}{\alpha + \tau} u + \frac{\tau}{\alpha + \tau} v \right).$$

### B.6 RUNTIME COMPARISON BETWEEN PROSPECT AND PSG-SRM

The per-epoch CPU runtimes for PROSPECT and PSG-SRM are reported in Table A2. These experiments are conducted on a MacBook Pro equipped with an Apple M1 Pro processor and a 16-GB RAM. The results indicate that PSG-SRM generally achieves faster executions than PROSPECT. The total wall-clock times of PROSPECT, SOREL, and PSG-SRM for directly solving the doubly-regularized SRM problem (5) are provided in Table A3. PSG-SRM runs the fastest across different cases, which indicates that it is effective in solving this problem.

### B.7 SUBOPTIMALITY EXPERIMENT

We follow (Mehta et al., 2024) to conduct the suboptimality experiment. The corresponding objective functions $F(w)$ for PSG-SRM, PROSPECT, and SOREL are given in (5), (3), and (4), respectively. The number of passes $k$ is defined as the number of calls of a single-sample oracle $(\ell_i(w), \nabla \ell_i(w))$ divided by $n$ (i.e., the size of training set). The suboptimality w.r.t. $k$ is calculated by

$$\texttt{Suboptimality}(w_k) = \frac{F(w_k) - F(w^*)}{F(w_0) - F(w^*)},$$

where $w^*$ is an approximated solution solved by LBFGS (Nocedal & Wright, 2006). To make a fair comparison, the same $\ell_1$ regularization $g(w) = \lambda\|w\|_1$ is used for both PSG-SRM and SOREL. PROSPECT can be seen as an ablated version of PSG-SRM by setting the NDR as zero. However, PROSPECT uses the differentiable regularization $\lambda\|w\|_2^2/2$ (because it cannot handle $\lambda\|w\|_1$), while PSG-SRM uses the non-differentiable regularization $\lambda\|w\|_1$. Since $\lambda\|w\|_2^2/2$ has better geometrical properties than $\lambda\|w\|_1$, this grants more advantage to PROSPECT against PSG-SRM in the suboptimality experiment. Results in Figure A1 show that PSG-SRM is better than SOREL in most cases, and competitive with PROSPECT in the overall performance. Besides, PSG-SRM is the fastest method that converges to flat plots in most cases.

Table A2: Per-epoch CPU runtimes of PROSPECT and PSG-SRM. The runtime is measured in seconds and reported in the form of mean± STD. The number of repetitions is twice the number of epochs.

| Data Set | $n_{train}$ | SRM | PROSPECT | PSG-SRM |
|---|---|---|---|---|
| yacht | 244 | CVaR | 0.049±0.001 | **0.048**±0.000 |
| | | Extremile | 0.063±0.001 | **0.062**±0.008 |
| | | ESRM | 0.063±0.000 | **0.060**±0.000 |
| energy | 614 | CVaR | 0.134±0.003 | **0.133**±0.003 |
| | | Extremile | 0.228±0.012 | **0.224**±0.002 |
| | | ESRM | 0.226±0.012 | **0.224**±0.015 |
| concrete | 824 | CVaR | 0.190±0.009 | **0.187**±0.006 |
| | | Extremile | 0.355±0.012 | **0.351**±0.012 |
| | | ESRM | 0.342±0.004 | **0.338**±0.012 |
| kin8nm | 6553 | CVaR | 11.701±0.249 | **11.623**±0.100 |
| | | Extremile | 16.178±0.126 | **16.175**±0.131 |
| | | ESRM | **15.045**±0.400 | 15.140±0.428 |
| power | 7654 | CVaR | 16.644±0.322 | **16.622**±0.296 |
| | | Extremile | 22.569±0.175 | **22.511**±0.164 |
| | | ESRM | 22.410±0.747 | **22.339**±0.758 |

Table A3: Total wall-clock times of PROSPECT, SOREL and PSG-SRM. The runtime is measured in seconds and reported in the form of mean± STD (10 repetitions).

| Data Set | $n_{train}$ | SRM | PROSPECT | SOREL | PSG-SRM |
|---|---|---|---|---|---|
| yacht | 244 | CVaR | 10.211±0.210 | 34.720±0.334 | **6.364**±0.188 |
| | | Extremile | 8.264±0.132 | 35.170±0.616 | **6.397**±0.133 |
| | | ESRM | 11.135±0.166 | 34.765±0.484 | **6.358**±0.113 |
| energy | 614 | CVaR | 26.861±1.006 | 86.385±1.582 | **18.093**± 0.812 |
| | | Extremile | 34.143±1.812 | 86.236± 1.783 | **17.608**±0.859 |
| | | ESRM | 33.725±1.805 | 85.553±1.828 | **17.580**±1.188 |
| concrete | 824 | CVaR | 36.908±1.763 | 115.108±2.757 | **25.596**±0.829 |
| | | Extremile | 50.368±1.716 | 114.107±2.969 | **25.085**±0.865 |
| | | ESRM | 50.148±1.591 | 114.559±3.137 | **24.857**±0.890 |
| kin8nm | 6553 | CVaR | 695.67±14.925 | 454.866±9.981 | **333.102**±8.892 |
| | | Extremile | 891.789±17.002 | 459.553±9.786 | **331.035**±8.705 |
| | | ESRM | 846.777±20.697 | 458.549±10.187 | **331.815**±9.465 |
| power | 7654 | CVaR | 955.973±18.645 | 533.611±9.667 | **436.830**±7.849 |
| | | Extremile | 1493.354±14.674 | 532.237±5.945 | **435.958**±4.205 |
| | | ESRM | 1630.118±24.780 | 533.489±12.415 | **436.108**±13.932 |

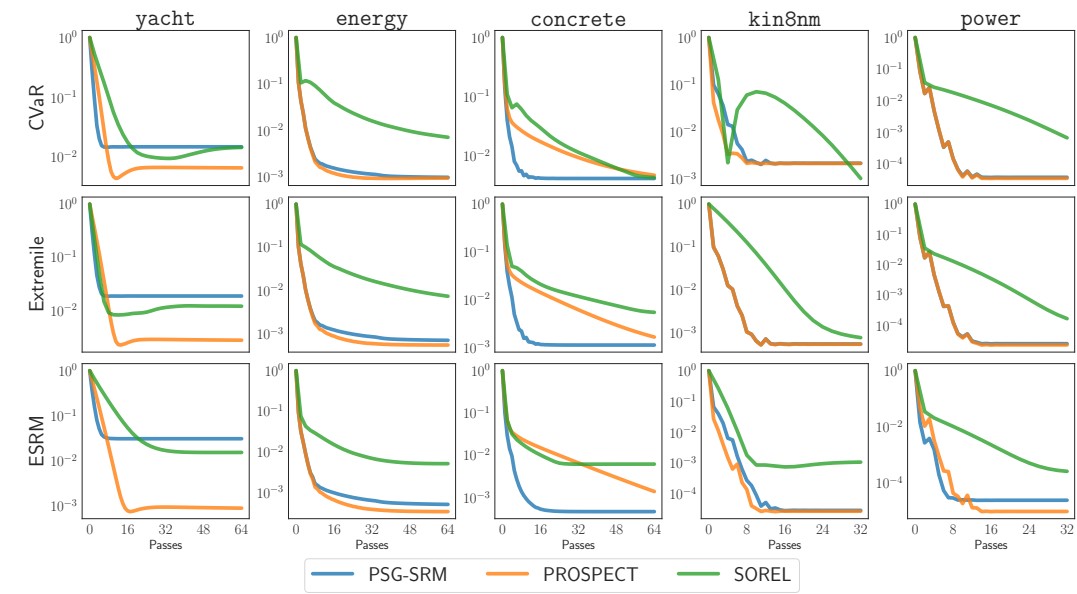

Figure A1: Suboptimality of PSG-SRM, PROSPECT, and SOREL on the regression benchmarks. The $y$-axis represents the suboptimality and the $x$-axis represents the number of passes.

## B.8 ROBUSTNESS TO HYPERPARAMETER CHANGE

Table A4 shows the robustness of PSG-SRM to hyperparameter change on `power`. For the convenience of demonstration, the regularization parameter is fixed at $\lambda = 1$, while the first row and the first column in each table contain different values of hyperparameters $\nu$ (shift cost) and $\eta$ (learning rate) for grid search, respectively. Results indicate that PSG-SRM achieves stable performance as the learning rate $\eta$ increases to a sufficiently large level $(0.01 \sim 0.03)$ across different values of shift cost $\nu$. Hence PSG-SRM is robust to certain levels of hyperparameter change.

Table A4: Performance of PSG-SRM with respect to different hyperparamters.

| SRM | $\eta$ \ $\nu$ | 0.01 | 0.1 | 1 | 10 | 100 |
|---|---|---|---|---|---|---|
| CVaR | $1 \times 10^{-5}$ | 0.4950 | 0.4950 | 0.4950 | 0.4950 | 0.4950 |
| | $3 \times 10^{-5}$ | 0.4950 | 0.4950 | 0.4950 | 0.4950 | 0.4950 |
| | $1 \times 10^{-4}$ | 0.4950 | 0.4950 | 0.4950 | 0.4950 | 0.4950 |
| | $3 \times 10^{-4}$ | 0.1011 | 0.1116 | 0.1725 | 0.2107 | 0.2166 |
| | $1 \times 10^{-3}$ | 0.0480 | 0.0503 | 0.0568 | 0.0586 | 0.0588 |
| | $3 \times 10^{-3}$ | 0.0389 | 0.0402 | 0.0434 | 0.0441 | 0.0442 |
| | 0.01 | 0.0370 | 0.0371 | 0.0374 | 0.0375 | 0.0375 |
| | 0.03 | 0.0369 | 0.0368 | 0.0369 | 0.0370 | 0.0370 |
| Extremile | $1 \times 10^{-5}$ | 0.4950 | 0.4950 | 0.4950 | 0.4950 | 0.4950 |
| | $3 \times 10^{-5}$ | 0.4950 | 0.4950 | 0.4950 | 0.4950 | 0.4950 |
| | $1 \times 10^{-4}$ | 0.4950 | 0.4950 | 0.4950 | 0.4950 | 0.4950 |
| | $3 \times 10^{-4}$ | 0.1230 | 0.1237 | 0.1725 | 0.2107 | 0.2166 |
| | $1 \times 10^{-3}$ | 0.0496 | 0.0505 | 0.0568 | 0.0586 | 0.0588 |
| | $3 \times 10^{-3}$ | 0.0395 | 0.0402 | 0.0434 | 0.0441 | 0.0442 |
| | 0.01 | 0.0370 | 0.0371 | 0.0374 | 0.0375 | 0.0375 |
| | 0.03 | 0.0369 | 0.0368 | 0.0369 | 0.0370 | 0.0370 |
| ESRM | $1 \times 10^{-5}$ | 0.4950 | 0.4950 | 0.4950 | 0.4950 | 0.4950 |
| | $3 \times 10^{-5}$ | 0.4950 | 0.4950 | 0.4950 | 0.4950 | 0.4950 |
| | $1 \times 10^{-4}$ | 0.4950 | 0.4950 | 0.4950 | 0.4950 | 0.4950 |
| | $3 \times 10^{-4}$ | 0.1566 | 0.1566 | 0.1738 | 0.2107 | 0.2166 |
| | $1 \times 10^{-3}$ | 0.0524 | 0.0525 | 0.0567 | 0.0586 | 0.0588 |
| | $3 \times 10^{-3}$ | 0.0409 | 0.0410 | 0.0433 | 0.0441 | 0.0442 |
| | 0.01 | 0.0372 | 0.0372 | 0.0374 | 0.0375 | 0.0375 |
| | 0.03 | 0.0369 | 0.0369 | 0.0369 | 0.0370 | 0.0370 |

