# OpenReview forum: "A Proximal Stochastic Gradient Method for Doubly-regularized Spectral Risk Minimization"
_ICLR.cc/2026/Conference — Submitted to ICLR 2026_

### Official Review · Reviewer_4ofy · 2025-10-24

**Soundness:** 3
**Presentation:** 2
**Contribution:** 3
**Rating:** 6
**Confidence:** 2

**Summary:**

This paper studies the Spectral risk minimization (SRM) problem with double regularizations: distribution shift regularization (DSR) and  non-differentiable regularization (NDR). A new method, proximal stochastic gradient method (PSG-SRM), is proposed to solve the problem and both theoretical and empirical results are shown.

**Strengths:**

1. This paper solves the challenging double regularization problems. With the non-differentiable regularization, it is challenging to apply the existing bias and variance analysis skills to the problem. To solve this problem, this paper proposes a new method, PSG-SRM, to solve the problem.

2. The authors provide the theoretical analysis for the proposed algorithm, which achieves  lower computational complexity than the two state-of-the-art methods that handle DSR or NDR separately.  Also the numerical results show that the proposed algorithm has good empirical performance.

**Weaknesses:**

1. The presentation of this paper is unclear and difficult to follow. The introduction contains too many equations and definitions, which would be more appropriately placed in the Preliminaries section. Additionally, there are extensive discussions and comparisons with existing works on objective formulations and convergence rates; it would be clearer to summarize these in a table for better illustration.

2. Some important definitions, such as those on lines 51 and 193, are located in the appendix. Since the Lipschitz property plays a crucial role in the theoretical proofs and lemmas, its definition should be moved to the main text rather than being placed in the appendix.

**Questions:**

While the main issue of this paper lies in its presentation, I also have a few technical concerns:

1. The paper claims that “standard proximal (stochastic) gradient methods like SOREL can handle NDRs but cannot handle DSR.” However, in Section 2.3, SOREL formulates the problem as a minimax optimization. Generally, with DSR regularization, the objective should be concave in  q, which would typically make the problem easier to solve rather than more difficult.

2. It would be helpful if the authors could provide a table summarizing the assumptions used in SOREL, PROSPEC, and the proposed method. Additionally, please clarify whether achieving similar linear convergence in this paper incurs any extra computational cost compared to PROSPEC.

3. From an algorithmic standpoint, is there any substantial difference between the proposed PSG-SRM and PROSPEC, aside from the proximal stochastic gradient step on  w?

---

> ### Author Response · Authors · 2025-11-27
>
> We thank the reviewer for the valuable feedback on our paper. We are pleased that the reviewer finds the paper sound and the contribution good. We acknowledge the concerns regarding the clarity of the presentation and have revised the manuscript to address all the points.
>
> **Q1.** Please note that the SOREL model in Section 2.3 is a minimax optimization that consists of an inner maximization and an outer minimization. **If one adds an DSR in it, this only eases the inner maximization, but consequently makes the outer minimization intractable: there is an NDR $g(w)$ in the outer minimization, which cannot be simultaneously handled along with the DSR by the existing SOREL algorithm.** This problem is exactly we intend to solve in this work.
>
> In fact, it is the proposed PSG-SRM that can solve a broader class of problems than SOREL. Since PSG-SRM can solve the doubly-regularized SRM problem in Eq. 5, one can simply set the shift cost $\nu=0$ in Eq. 5 and then it is reduced to the single NDR problem of SOREL.
>
> **Q2.** We add Table 1 in Section 1 to compare the three algorithms for SRM, including their assumptions, functions, and computational complexities. PSG-SRM solves the most general problem with the lowest computational complexity among the three.
>
> Achieving linear convergence with PSG-SRM does not incur an extra computational cost. In fact, it is shown in Table 1 to have a lower complexity than PROSPECT. The complexity for PROSPECT has a quadratic dependence on the number of training samples $n$ in its nominator ($C'_4 n^2$), whereas the constant term for PSG-SRM only depends linearly on $n$ ($C_1 \mu n$). When $n$ is large, this difference means that PSG-SRM achieves the same convergence tolerance $\varepsilon$ with a lower computational complexity than PROSPECT. This theoretical advantage is also supported by the empirical wall-clock time comparisons in the newly-added Table A3 of the revised manuscript.
>
> **Q3.** The difference is substantial and lies in the methodology for integrating the NDR $g(w)$ into the variance reduction framework.
>
> * Limitation of PROSPECT: PROSPECT uses a variance reduction technique that relies on the differentiability of the entire objective function to absorb the regularization term into the stochastic gradient estimate. This technique fundamentally fails when the regularizer $g(w)$ is an NDR, such as the commonly used $\ell_1$ regularization.
>
> * Novelty of PSG-SRM: PSG-SRM is not just PROSPECT with a proximal step. We introduce the $\text{prox}_{\eta g}(\cdot)$ operator to skillfully separate the non-differentiable term $g$ from the differentiable risk function $f$. The key methodological contribution is to show that the non-expansiveness of the proximal mapping can be exploited to preserve the existing bias and variance reduction mechanism. This leads to a novel convergence analysis, captured by the descent property (Lemma 4) and a new Lyapunov function (Theorem 1). It ensures that we can retain the linear convergence rate while handling the NDR.
>
> **W1.** We clarify the development path for the field of stochastic gradient methods for SRM in the paragraph below Eq. 5 in the revised manuscript: (1) DSR $\to$ (2) SRM+DSR $\to$ (3) SRM+DSR with strongly convex loss $\to$ (4) SRM+NDR with strongly convex regularizer $\to$ (5) SRM+DSR+NDR (ours) for Eqs. 1 to 5. We also add Table 1 in Section 1 to compare the three algorithms for SRM, including their assumptions, functions, and computational complexities.
>
> **W2.** Line 51 actually indicates maintaining a table of losses $l\approx \ell(w)$ for the current iterate $w$. The reason for doing this is rather subtle and lengthy, thus put in Appendix A.3.
>
> Other important terms like Lipschitz properties are defined in Eqs. 11 to 13 of the revised manuscript. The rest minor terms are explained in Appendix A.1.

---

### Official Review · Reviewer_HVyX · 2025-10-30

**Soundness:** 3
**Presentation:** 3
**Contribution:** 2
**Rating:** 4
**Confidence:** 4

**Summary:**

This paper proposes PSG-SRM to solve distribution shift regularization (DSR) problems with nonsmooth regularization terms. Unlike previous methods, the proposed approach can simultaneously handle problems that involve both a nonsmooth convex regularizer and distribution shift regularization. The authors prove the linear convergence of the algorithm theoretically. Experimental results demonstrate that the algorithm exhibits stronger robustness than other methods in most cases.

**Strengths:**

This paper introduces an interesting problem related to DRO and fairness machine learning. It is well written and easy to follow. The authros  develop an Lyapunov function to prove the convergence of their proposed algorithm. The empirical validation uses a reasonable diversity of datasets and kinds of spectral risks.

**Weaknesses:**

1. The proposed algorithm is mainly based on Prospect, and it seems that the only modification is replacing the gradient descent on parameter $w$ with proximal gradient descent.

2. The authors should clearly specify the assumptions used in each Lemma and Theorem.

3. In Line 192, the authors assume that each loss function is both strongly convex and Lipschitz continuous. However, these two assumptions cannot hold simultaneously unless the function is restricted to a compact set.

4. In Line 325, the authors state that the proximal gradient method can only achieve a sublinear convergence rate because the regularizer is nonsmooth or non-contractive. However, in [1], the authors study the spectral risk minimization problem without distribution shift regularization, which is a strongly convex–concave minimax problem. The author in [1] claims that the lower complexity bound is sublinear. Could the authors further clarify this point?

5. The authors claim that their algorithm has an advantage in runtime. However, according to Table A.2, its runtime does not show a significant improvement compared to Prospect.


Reference

[1] Ge, Y., & Jiang, R. (2024). SOREL: A Stochastic Algorithm for Spectral Risks Minimization. International Conference on Learning Representations, abs/2407.14618. https://doi.org/10.48550/arXiv.2407.14618

**Questions:**

See weaknesses.

---

> ### Author Response · Authors · 2025-11-27
>
> We appreciate the reviewer's thorough review, positive summary, and valuable feedback, which will help us improve the clarity and rigor of this work. We are pleased that the reviewer finds the problem interesting, the paper well-written, and the use of the Lyapunov function for convergence analysis to be a strength.
>
> **W1.** While PSG-SRM incorporates the core bias and variance reduction mechanism from PROSPECT, the transition to handle Non-Differentiable Regularization (NDR) $g(w)$ is a non-trivial and essential contribution, as PROSPECT is fundamentally unable to solve the doubly-regularized model Eq. 5. The variance reduction scheme in PROSPECT requires the regularizer to be differentiable, so that its gradient can be absorbed into the loss gradient $\nabla \ell(w)$. For an NDR, $\nabla g(w)$ does not exist, and thus PROSPECT is unavailable.
>
> Our central methodological challenge is to introduce the proximal step for $g(w)$ while preserving the effectiveness of the variance reduction mechanism. We achieve this by skillfully exploiting the non-expansiveness of the proximal mapping ($\text{prox}_{\eta g}$) to avoid expanding the bias and variance of the stochastic gradient gap $\Delta^{(t)}$.
>
> This new scheme requires developing a new descent property (Lemma 4) and a completely new Lyapunov function (Theorem 1) to establish linear convergence, which is more than a simple substitution. It is a novel integration that solves a more complex problem while maintaining a superior convergence rate. Its realization spans throughout Appendices A.8 and A.9, involving extensive subtle work.
>
> **W2.** To keep concise in presenting the lemmas and theorems, we put the primary assumptions for them at the beginning of Section 3. Specifically, Each $\ell_i(w)$ is $\mu$-strongly convex, $G$-Lipschitz continuous (function value), and $L_{\ell}$-smooth (gradient). $g(w)$ is a proper, lower semicontinuous, and (not necessarily strongly) convex NDR. Besides, extra assumptions or conditions for each specific lemma or theorem are presented individually.
>
> **W3.** The condition that "each loss function is both strongly convex and Lipschitz continuous" is necessary and widely-used in this field, including the two baselines PROSPECT and SOREL. In fact, it can be generally satisfied in practical implementation. Without loss of generality, suppose the optimization starts at $w^{0}$. Then a necessary condition of optimizing tractability is the level-boundedness of the entire objective function: $\{w:F(w)\leqslant  F(w^{0}) \}$ is bounded ref. (Rockafellar \& Uryasev, 2000), otherwise $w^{t}$ can tend to infinity even if $F(w^{t})$ descends. In a $d$-dimensional real space, this bounded set is also compact. Then based on the descent property Lemma 3, the subsequent level sets $\{w:F(w)\leqslant  F(w^{t}) \}_t$ are all subsets of the initial level set $\{w:F(w)\leqslant  F(w^{0}) \}$, which are all compact. This satisfies the compact set restriction indicated by the reviewer.
>
> **W4.** We double check the corresponding statements in SOREL ref. (Ge \& Jiang, 2025). In the "Our Contributions" paragraph on Page 2, the authors of SOREL claim that "SOREL achieves a near-optimal rate of $\tilde{O}(1/\sqrt{\epsilon})$, which matches the known lower bound of $\Omega(1/\sqrt{\epsilon})$" **in the deterministic settingref. (Ouyang \& Xu, 2021)** . Therefore, the lower complexity bound you referred to is actually from the **deterministic setting of ref. (Ouyang \& Xu, 2021)**, not from the stochastic setting from SOREL ref. (Ge \& Jiang, 2025).
>
> **W5.** In Table A2, PROSPECT uses the differentiable regularization $\lambda \\|w\\|_2^2/2$, while PSG-SRM uses the non-differentiable regularization $\lambda\\|w\\|_1$. Since $\lambda \\|w\\|_2^2/2$ has better geometrical properties than $\lambda\\|w\\|_1$, this grants more advantage to PROSPECT against PSG-SRM, but PSG-SRM still outperforms PROSPECT in most cases. In this revision, we add the total wall-clock times of PROSPECT, SOREL, and PSG-SRM for directly solving the doubly-regularized SRM problem Eq. 5 in Table A3. Results show that PSG-SRM achieves significant advantage over PROSPECT and SOREL in solving Eq. 5.
>
> Reference:
>
> Yuyuan Ouyang and Yangyang Xu. Lower complexity bounds of first-order methods for convex concave
> bilinear saddle-point problems. Mathematical Programming, 185(1-2):1–35, 2021.

---

### Official Review · Reviewer_JjeZ · 2025-10-31

**Soundness:** 3
**Presentation:** 2
**Contribution:** 2
**Rating:** 4
**Confidence:** 4

**Summary:**

This paper proposes PSG-SRM, a novel proximal stochastic gradient method for optimizing spectral risk measures. The algorithm incorporates double regularization, simultaneously reduces bias and variance over iterations, and achieves linear convergence under appropriate assumptions. Compared to the prior work PROSPECT, PSG-SRM enjoys lower per-iteration computational complexity, making it theoretically more scalable to large datasets. Empirical results demonstrate that PSG-SRM delivers consistently better and more stable optimization performance.

**Strengths:**

1. The paper provides a rigorous theoretical analysis showing that PSG-SRM achieves linear convergence, and clearly establishes its lower computational complexity compared to PROSPECT.

2. The experimental evaluation is conducted on widely accepted benchmarks in the field, and effectively demonstrates that PSG-SRM attains superior and more stable optimization performance in practice.

**Weaknesses:**

1. Readability: The writing suffers from several issues, including:
1.1 Excessive use of abbreviations (e.g., SRM, DSR, NDR, DRO, PAV) without sufficient introduction or justification;
1.2 Confusing mathematical notation (e.g., Equation 5 uses notation for DSR and NDR that appears as if it belongs in a denominator, which is misleading);
1.3 Arbitrary use of boldface text, which disrupts the flow and professionalism of the presentation.

2. Mismatch between theory and experiments: The theoretical contribution centers on faster convergence (linear rate) and reduced computational complexity relative to PROSPECT. However, the experiments only report final performance metrics and do not validate the claimed convergence speed or iteration complexity. For instance, there is no plot showing loss vs. epoch or wall-clock time.

3. Weak motivation for the core formulation: The key contribution (Equation 5) appears to be a straightforward combination of PROSPECT’s objective with an additional strongly convex, non-differentiable regularization. The explanation “Since NDRs are widely used to handle complicated tasks, …, it motivates us to unify NDR and DSR…” is overly heuristic. The authors should provide a deeper rationale for this specific unification.

**Questions:**

1. The paper theoretically establishes that PSG-SRM converges faster than existing methods. Can this be empirically verified? For example, does PSG-SRM reach the same objective value or test performance in fewer epochs or less wall-clock time?

2. The Introduction introduces five distinct optimization objectives, which feels overwhelming. In contrast, the cited baselines (PROSPECT, LSVRG, SOREL) typically frame their problems using only two objectives. Could the exposition be simplified to better highlight the specific problem PSG-SRM addresses?

3. Section 3.1 is dedicated solely to prove the smoothness of f. However, this property is standard in stochastic optimization, and the section does not appear to add novel insight. Is this section necessary?

---

> ### Author Response · Authors · 2025-11-27
>
> We thank the reviewer for summarizing the core contributions of our paper, especially acknowledging the rigorous theoretical analysis regarding linear convergence and the lower computational complexity compared to PROSPECT , as well as the superior empirical performance of PSG-SRM.
>
> **Q1 \& W2.** Experimental results on the least-square regression and out-of-distribution classification in Sections 4.2 and 4.3 show that PSG-SRM achieves the best performance in most cases. Figure A1 shows that PSG-SRM lands to a flat suboptimality plot faster than SOREL and PROSPECT in terms of epochs. Table A3 of the revised manuscript shows the total wall-clock times of PROSPECT, SOREL, and PSG-SRM, where PSG-SRM runs the fastest across different cases. These results empirically verify that the performance and convergence of PSG-SRM accord with its theory.
>
> **Q2.** We double check the five objectives and find that they show a clear development path that includes some remarkable cornerstones in this field: (1) DSR $\to$ (2) SRM+DSR $\to$ (3) SRM+DSR with strongly convex loss $\to$ (4) SRM+NDR with strongly convex regularizer $\to$ (5) SRM+DSR+NDR (ours) for Eqs. 1 $to$ 5. Removing any one of them may affect the understanding of the history of this field and the motivation of our work. We clarify this development path in the paragraph below Eq. 5 in the revised manuscript.
>
> **Q3.** While the existence of the property is standard, the explicit derivation and bounding of the Lipschitz constant $L_f$ (Lemma 1) is critical and non-standard in this specific context. The value $L_f:=M_q L_{\ell}+ \sqrt{n} G L_{q\circ\ell}$ is a compound constant, which depends on multiple terms. Moreover, This specific, explicitly-bounded value of $L_f$ is essential for setting the appropriate learning rate ($\eta=\frac{1}{(1+\delta)L_f}$) in Algorithm 1 and is a crucial parameter in the subsequent linear convergence proof (Theorem 1, not shown in the snippets but implied by the analysis in the paper).
>
> **W1.** We revise all abbreviations (SRM, DSR, NDR, DRO, PAV, etc.) on their first occurrence and ensure that they are used consistently. We have changed the single underlines in Eq. 5 into double underlines to better highlight that DSR and NDR are descriptions rather than denominators. We also revise the entire manuscript and significantly reduce the arbitrary use of boldface to improve the professional tone and flow of the paper.
>
> **W3.** We clarify that the motivation for Eq. 5 is not simply combining two existing regularizers, but resolving a critical trade-off faced by state-of-the-art methods:
>
> * PROSPECT (DSR-only) achieves distributional robustness and linear convergence but cannot handle NDR due to the nature of its variance reduction mechanism.
>
> * SOREL (NDR-only) handles an NDR, but only a strongly convex one, and by dropping the DSR, it loses distributional robustness and achieves only a sublinear convergence rate.
>
> The deep motivation for the doubly-regularized SRM model Eq. 5 lies in two main aspects:
>
> * Model Flexibility: It allows the use of widely-used, but non-strongly-convex NDRs, such as the $\ell_1$-regularization ($g(w) = \lambda \\|w\\|_1$). This important case can be accommodated by neither the PROSPECT-based approach nor the SOREL formulation Eq. 4.
>
> * Algorithmic Innovation: The main technical challenge and innovation lies in developing a mechanism that preserves the variance and bias reduction required for linear convergence (like PROSPECT) while simultaneously incorporating the non-expansive proximal mapping required for the non-differentiable NDR (like that in SOREL). This is far from a "straightforward combination". While the motivation is intuitive, the realization is rather difficult.
>
> The key technique is to skillfully exploit the full proximal gradient step $\bar{w}^{(t+1)}$ in Eq. 15, which is independent of $i_t$, to separate the stochastic terms from the deterministic terms in Eq. 20. The specific realization of this technique spans throughout Appendices A.8 and A.9, involving extensive subtle work.

---

### Official Review · Reviewer_7taK · 2025-10-31

**Soundness:** 3
**Presentation:** 3
**Contribution:** 2
**Rating:** 4
**Confidence:** 3

**Summary:**

This paper studies doubly-regularized spectral risk minimization (SRM) problems, which combine distribution-shift regularization (DSR) on the spectrum and a potentially non-differentiable parameter regularizer (NDR). The authors propose PSG-SRM, a proximal stochastic gradient algorithm that integrates a variance and bias control scheme (extended from PROSPECT-style estimators), leverages the non-expansiveness of proximal operators to handle NDR, and establishes linear convergence under the assumption that each sample loss $\ell_i$ is strongly convex. The paper claims superior computational complexity compared to methods that handle only DSR or NDR individually, and presents experimental results on regression and classification tasks showing competitive and stable performance.

**Strengths:**

1. The paper addresses a practically relevant problem by integrating both distributional robustness and nonsmooth regularization, filling a gap in existing methodology.
2. Theoretical contributions include lemmas bounding bias and variance and the construction of a Lyapunov function to establish linear convergence.
3. Experiments cover regression and classification tasks with multiple baselines, demonstrating stable performance across settings.

**Weaknesses:**

1. The proposed algorithm is incremental compared to PROSPECT, with its main novelty lying in the proximal stochastic gradient step.
2. On Page 4, Line 191, the authors assume that each $\ell_i(w)$ is $\mu$-strongly convex, $G$-Lipschitz continuous, and $L_\ell$-smooth. These conditions are often violated in real-world loss functions. The paper should more clearly discuss the practical regimes where these assumptions hold.
3. In Table 3, the evaluation of out-of-distribution (OOD) performance uses inconsistent metrics: "worst group test error" is reported for the 'amazon' dataset, while "median group test error" is used for 'iwildcam'. This selective reporting impedes direct and rigorous comparison and may suggest cherry-picking of favorable metrics.

**Questions:**

See the weakness.

---

> ### Author Response · Authors · 2025-11-27
>
> We sincerely thank the reviewer for the time and effort in reviewing our manuscript, as well as for the constructive and encouraging feedback. We are pleased that the reviewer finds the paper to be sound and well-presented, and we appreciate the recognition of the practical relevance, the novelty of integrating distribution shift regularization (DSR) and non-differentiable regularization (NDR), the strong theoretical contributions (bias/variance bounds and linear convergence proof), and the comprehensive experimental results.
>
> **W1.** We respectfully disagree with the assessment of incrementality. While this work builds upon the foundational framework of variance-reduced stochastic gradient methods (like PROSPECT), the introduction of the NDR poses a significant challenge that requires a non-trivial methodological and theoretical extension.
>
> * Novel Problem Setting: Existing works like PROSPECT and SOREL focus only on either DSR or NDR, instead of both of them. Our work is the first to propose an algorithm and provide theoretical analysis for the doubly-regularized SRM problem, which simultaneously handles both DSR and NDR. This combined setting is crucial for practical applications where nonsmooth regularizers (e.g., $\ell_1$ for sparsity) are desired alongside distributional robustness.
>
> * Non-Trivial Extension: The main novelty is not just the proximal step, but the integration of the non-expansive proximal operator into the PROSPECT-style bias and variance control scheme, and the subsequent construction of a Lyapunov function to establish linear convergence for the proposed PSG-SRM. Extending the convergence proof to handle the non-smooth term requires significant theoretical modification of the original analysis. This is a complete solution for a fundamentally more general and complex problem class. The specific realization of this technique spans throughout Appendices A.8 and A.9, involving extensive subtle work.
>
> **W2.** We add Table 1 in the revised manuscript, which shows that these conditions are necessary and shared in this field (stochastic gradient methods for SRM). They are primarily used to establish the linear convergence rate of the algorithm. They are standard requirements for proofs of linear convergence in variance-reduced stochastic optimization methods (e.g., SAGA, Prox-SAGA, and their extensions like PROSPECT and SOREL).
>
> We clarify that these assumptions hold in several practical scenarios, such as linear regression with a strongly convex objective function (e.g., when an $\ell_2$ regularizer is part of the loss), or in classification problems when a sufficiently strong $\ell_2$ regularization term is added to the objective. Besides, such theoretical analysis is necessary and elemental to reveal properties in a locally convex setting, which build up the entire algorithmic properties in various general and practical scenarios.
>
> **W3.** First, we strictly follow the evaluating criterion of PROSPECT and SOREL to report experimental results for these two data sets, which is not an attempt at selective reporting. Second, the amazon data set has $5$ groups, while the iwildcam data set has $60$ groups. Since the latter has significantly more groups, it is reasonable to use the worst group error and the median group error for the former and the latter, respectively. If the worst group error is used for iwildcam, the corresponding result will be extremely biased by only one out of $60$ groups, which cannot reflect the overall performance of a method. Therefore, the median group error is a robust choice for these competitors.

---

### Author Response · Authors · 2025-12-02
**Summary of Discussions [2/2]**

## 2. Theoretical Assumptions and Practical Regimes (W2 for `7taK`, W3 for `HVyX`) ##

The assumptions that each sample loss is $\mu$-strongly convex, $G$-Lipschitz continuous, and $L_{\ell}$-smooth are **necessary and standard requirements for proofs of linear convergence in variance-reduced stochastic optimization methods like PROSPECT and SOREL**.

* **Practicality:** We clarify that these assumptions hold in practical scenarios such as linear regression with a strongly convex objective (e.g., with an $\ell_2$ regularizer) or in classification problems with sufficient $\ell_2$ regularization.

* **Compact Set:** We clarify that the necessary condition of level-boundedness of the objective function, combined with the descent property, naturally confines the subsequent level sets to compact sets, which satisfies the restriction mentioned by reviewer `HVyX`.

## 3. Empirical Verification of Convergence Speed (W2 \& Q1 for `JjeZ`, W5 for `HVyX`, Q2 for `4ofy`) ##

The revised manuscript now includes explicit empirical evidence that validates the theoretical advantage in convergence speed and runtime:

* **Convergence Plot:** Figure A1 shows that PSG-SRM achieves a flat suboptimality plot faster than SOREL and PROSPECT in terms of **epochs**.

* **Wall-Clock Time:** The new **Table A3** shows the total wall-clock times of the three algorithms, which indicate that **PSG-SRM runs the fastest** across different cases. It confirms the theoretical advantage of PSG-SRM in computational complexity.

## 4. Presentation Issues (W1 for `JjeZ`, W3 for `7taK`, W1 \& Q2 for `4ofy`) ##

We have made significant revisions to improve the presentation and readability of manuscript:

* **Clarity:** We define all abbreviations (SRM, DSR, NDR, DRO, PAV) on their first occurrence. We have reduced the arbitrary use of boldface text.

* **Equation Notation:** We change the single underlines in Eq. 5 to double underlines to clarify that DSR and NDR are descriptions, not denominators.

* **Objectives:** We clarify the development path of the five objectives (Eqs. 1 to 5) to better motivate the doubly-regularized model, because they represent crucial cornerstones in this field.

* **Comparative Table:** We add **Table 1** in Section 1 to summarize the assumptions, functions, and computational complexities of PROSPECT, SOREL, and PSG-SRM, which provides a clearer overview.

* **Evaluating Metrics:** We clarify that the use of "worst group test error" for the 'amazon' data set and "median group test error" for 'iwildcam' is **not selective reporting but strictly follows the criterion** of SOREL and PROSPECT. This is a necessary measure because iwildcam has significantly more groups than amazon ($60$ vs. $5$). Using the worst-case error for iwildcam will be extremely biased by a single group and cannot reflect the overall performance of a method.

---

### Author Response · Authors · 2025-12-02
**Summary of Discussions [1/2]**

We sincerely thank the reviewers for their time, effort, and constructive feedback on our manuscript. We are pleased that the reviewers find our work to be **well-written** [`HVyX`], and to present a **practically relevant** [`7taK`] problem. Our core contributions—integrating **Distribution-Shift Regularization (DSR)** and **Non-Differentiable Regularization (NDR)** into the **doubly-regularized** Spectral Risk Minimization (SRM) problem, the **rigorous theoretical analysis** (including bias/variance bounds and linear convergence proof), and the **superior empirical performance of PSG-SRM (Proximal Stochastic Gradient SRM)**—have been recognized. [`7taK`: _The paper addresses a practically relevant problem by integrating both distributional robustness and nonsmooth regularization, filling a gap in existing methodology_. `JjeZ`: _The paper provides a rigorous theoretical analysis showing that PSG-SRM achieves linear convergence, and clearly establishes its lower computational complexity compared to PROSPECT_. `HVyX`: _Unlike previous methods, the proposed approach can simultaneously handle problems that involve both a nonsmooth convex regularizer and distribution shift regularization_. `4ofy`: _This paper solves the challenging double regularization problems. With the non-differentiable regularization, it is challenging to apply the existing bias and variance analysis skills to the problem. To solve this problem, this paper proposes a new method, PSG-SRM, to solve the problem_. ]

# Core Contributions Acknowledged #

* **Novel Problem Setting:** This work is the **first** to propose an algorithm and analysis for the **doubly-regularized** SRM problem, which simultaneously handles **both DSR and NDR**. [`7taK`: _The paper addresses a practically relevant problem by integrating both distributional robustness and nonsmooth regularization, filling a gap in existing methodology_. `JjeZ`: _The algorithm incorporates double regularization, simultaneously reduces bias and variance over iterations_. `HVyX`: _Unlike previous methods, the proposed approach can simultaneously handle problems that involve both a nonsmooth convex regularizer and distribution shift regularization_. `4ofy`: _This paper solves the challenging double regularization problems_. ]

* **Algorithmic Superiority:** The proposed **PSG-SRM** method achieves **linear convergence** and is theoretically more scalable due to its **lower computational complexity** compared to the DSR-only baseline PROSPECT and the NDR-only baseline SOREL (**see Tables 1 and A3**). [`7taK`: _establish linear convergence_. `JjeZ`: _PSG-SRM achieves linear convergence, and clearly establishes its lower computational complexity compared to PROSPECT_. `HVyX`: _The authors prove the linear convergence of the algorithm theoretically_. `4ofy`: _The authors provide the theoretical analysis for the proposed algorithm, which achieves lower computational complexity than the two state-of-the-art methods that handle DSR or NDR separately_. ]

# Clarifications and Revisions to Address Main Concerns #

## 1. Incrementality and Core Novelty (W1 for `7taK`, W3 for `JjeZ`, W1 for `HVyX`) ##

We respectfully disagree with the assessment of incrementality. While PSG-SRM stands on the variance reduction framework of PROSPECT, the **integration of NDR is a nontrivial methodological and theoretical extension**.

* **Limitation of PROSPECT:** PROSPECT cannot fundamentally solve the doubly-regularized model because its variance reduction scheme relies on the **differentiability** of the regularizer, which is violated by an NDR (e.g., $\ell_1$ regularization).

* **Innovation of PSG-SRM:** Our key novelty lies in **skillfully exploiting the non-expansiveness of the proximal mapping** to separate the non-differentiable term from the differentiable risk function, in order to preserve the variance and bias reduction mechanism required for linear convergence. This requires a **new descent property (Lemma 4)** and a **completely new Lyapunov function (Theorem 1)**. It is a complete solution for a fundamentally more complex problem class.

---

### Meta-Review · Area_Chair_5B3F · 2025-12-23

**Summary:**

This paper proposes a new algorithm for doubly-regularized spectral risk minimization. The reviewers pointed out a few concerns about the paper, including the assumptions are unrealistic, the algorithm is an incremental/slight modification of existing one, numerical experimental results are not convincing. Among these issues, the most fundamental one is about the assumptions. For example, reviewer 7taK pointed out that “These conditions are often violated in real-world loss functions. The paper should more clearly discuss the practical regimes where these assumptions hold.” In fact, the assumptions presented in equations (11)-(12) can’t hold simultaneously. That is, one can’t assume that a function is simultaneously strongly convex and Lipschitz continuous in the whole space. This has been pointed out in the literature, for example, see the first paragraph on page 1355 of “B. Grimmer, Convergence Rates for Deterministic and Stochastic Subgradient Methods without Lipschitz Continuity, SIOPT, 29(2): 1350-1365, 2019”. The authors are urged to carefully check and correct this when the paper is submitted to a future venue.

**Reviewer Concerns:**

The following concerns are still outstanding:
1. The method and contribution are incremental.
2. Assumptions are problematic.

**Reviewer Scores:**

There are fundamental flaws on the assumptions, and the reviewers who questioned on this are not likely to increase their scores.

---

### Decision · Program_Chairs · 2026-01-26

Reject